# The HSF1–PARP13–PARP1 complex facilitates DNA repair and promotes mammary tumorigenesis

Mitsuaki Fujimoto[1], Ryosuke Takii[1], Eiichi Takaki[1], Arpit Katiyar[1], Ryuichiro Nakato[2], Katsuhiko Shirahige[2] & Akira Nakai [1]

Poly(ADP-ribose) polymerase 1 (PARP1) is involved in DNA repair, chromatin structure, and transcription. However, the mechanisms that regulate PARP1 distribution on DNA are poorly understood. Here, we show that heat shock transcription factor 1 (HSF1) recruits PARP1 through the scaffold protein PARP13. In response to DNA damage, activated and auto-poly-ADP-ribosylated PARP1 dissociates from HSF1–PARP13, and redistributes to DNA lesions and DNA damage-inducible gene loci. Histone deacetylase 1 maintains PARP1 in the ternary complex by inactivating PARP1 through deacetylation. Blocking ternary complex formation impairs redistribution of PARP1 during DNA damage, which reduces gene expression and DNA repair. Furthermore, ternary complex formation and PARP1 redistribution protect cells from DNA damage by promoting DNA repair, and support growth of BRCA1-null mammary tumors, which are sensitive to PARP inhibitors. Our findings identify HSF1 as a regulator of genome integrity and define this function as a guarding mechanism for a specific type of mammary tumorigenesis.

---

[1] Department of Biochemistry and Molecular Biology, Yamaguchi University School of Medicine, Minami-Kogushi 1-1-1, Ube 755-8505, Japan. [2] Research Center for Epigenetic Disease, Institute of Molecular and Cellular Biosciences, The University of Tokyo, Tokyo 113-0032, Japan. Correspondence and requests for materials should be addressed to A.N. (email: anakai@yamaguchi-u.ac.jp)

Cellular homeostasis involves maintaining an intracellular balance of proteins and nucleic acids to keep a cell healthy. In order to cope with a variety of environmental and metabolic perturbations, cells have evolved sophisticated surveillance mechanisms including the DNA damage response (DDR) pathway to repair lesions in the DNA and facilitate replication[1, 2].

DDR proteins have an impact on a variety of cellular processes including DNA repair, chromatin remodeling, transcription, and cell cycle checkpoint. During DNA repair, signaling and repair proteins assemble at DNA lesions in a sequential and coordinated manner. Among these, poly(ADP-ribose) polymerase 1 (PARP1) is one of the first signaling proteins recruited to DNA breaks, including both single-strand breaks (SSBs)[3–5] and double-strand breaks (DSBs), which are repaired by two pathways: homologous recombination repair (HRR) and nonhomologous end-joining (NHEJ)[6, 7]. PARP1 facilitates the recruitment of DNA repair factors, such as RAD51 and 53BP1, chromatin remodeling

factors, and histone modifying emzymes to DNA lesions, and its deficiency results in reduced efficiency of HRR and NHEJ[6–9]. On the other hand, PARP1 also regulates transcription of inducible genes in response to stimuli such as heat shock and hormone treatment through poly(ADP-ribose) (PAR) modification of histones[10–14]. Importantly, the chromatin-related functions of PARP1 are associated with its redistribution to both DNA lesions and transcribed gene loci. However, the mechanisms of DNA damage-induced redistribution of PARP1 have not been elucidated in mammals.

To counteract protein misfolding, cells have also evolved mechanisms termed the proteotoxic stress response that adjusts proteostasis capacity or the buffering capacity for misfolded proteins through regulation of gene expression[15–17]. One universally conserved proteotoxic stress response is the heat shock response (HSR), which is characterized by induction of a small number of highly conserved heat shock proteins (HSPs or chaperones)[18, 19]. The HSR is mainly regulated at the level of transcription by an ancient transcription factor, heat shock factor (HSF), in eukaryotes. Among HSF family members (HSF1–HSF4) in mammals, HSF1 is a master regulator of the HSR. HSF1 mostly remains as an inert monomer in unstressed cells, and is converted to an active trimer that binds to the heat shock response element (HSE) and robustly induces the expression of HSPs during heat shock[20–22].

Even under unstressed conditions, HSF1 has a role in development and aging by regulating the expression of target genes including *HSP* and non-HSP genes, and HSF1 activity is tightly related with the progression of age-related neurodegenerative diseases[17, 23, 24]. HSF1 is also activated and supports growth of malignant tumors, in part by inhibiting aggregate formation and amyloidogenesis[25, 26]. Under physiological and pathological conditions, HSF1 activity is modulated by post-translational modifications including phosphorylation and acetylation[19, 24]. Recent genome-wide studies identified hundreds of constitutive HSF1-binding sites in immortalized and malignant tumor cells[27–30]. In fact, a small amount of the HSF1 trimer constitutively binds to nucleosomal DNA in complex with replication protein A and the histone chaperone FACT (facilitates chromatin transcription)[31, 32].

Here, we show that HSF1 and PARP1 form a complex through the scaffold protein PARP13. HSF1-dependent pre-recruitment of PARP1 on DNA is required for redistribution of PARP1 to DNA damage-inducible gene loci and DNA lesions during DNA damage. Furthermore, the HSF1-mediated DDR mechanisms protect tumor cells from DNA damage, especially supporting growth of BRCA1-null mammary tumors, which are sensitive to PARP inhibitors.

## Results

**HSF1 and PARP1 form a complex through the scaffold PARP13.** Because PARP13, which is also known as zinc finger antiviral protein (ZAP or ZC3HAV), was shown previously to be a human HSF1 (hHSF1)-interacting protein[32], we examined the interaction of hHSF1 with human PARPs including DNA-dependent PARPs (PARP1, 2), and RNA-binding CCCH-PARPs (PARP7, 12, 13)[33]. We found that HSF1 interacted with PARP1, PARP13, and a truncated isoform PARP13S[33] in cell extracts (Fig. 1a). Purified hPARP13-His directly interacted with both purified GST-hPARP1 and GST-hHSF1, but not with GST-hHSF2 or GST-hHSF4 in a GST pull-down assay (Fig. 1b). PARP1 and PARP13 (full-length and truncated PARP13) interacted with HSF1 in nuclear fractions (Fig. 1c). Furthermore, endogenous PARP13 interacted with HSF1 in the absence of PARP1, whereas PARP13 was required for the interaction of PARP1 with HSF1. Taken together, these results indicate that HSF1 and PARP1 form a complex through a scaffold protein PARP13.

PARP1 is an abundant nuclear protein and exerts a broad range of functions in the nucleus[3, 5], but it is not known whether PARP13 acts in the nucleus[34]. PARP13 localized mostly in the cytoplasm, but accumulated in the nucleus 3 h after treatment with a nuclear export inhibitor Leptomycin B (LMB) (Supplementary Fig. 1a, b), which suggests that PARP13 is shuttling between the nucleus and cytoplasm.

GST pull-down assay using hHSF1 deletion mutants showed that PARP13 interacted with the DNA-binding domain (DBD) of hHSF1 (Fig. 1d). To limit the interacting site, we substituted thirteen residues in hHSF1, which were different among the three hHSFs, with alanine or glutamic acid (Fig. 1e, black and white dots). Substitution of Thr20 or Ala33 abolished the interaction without affecting DNA-binding activity in vitro (Supplementary Fig. 1c, d). In addition, substitution of these residues with other amino acids including those found in hHSF2 or hHSF4 also abolished the interaction (Fig. 1f). Thus, PARP13 may in part contact Thr20 at the helix α1 (H1) in the winged helix-turn-helix motif as well as an adjacent Ala33, located on the surface of the DBD–DNA complex[35, 36]. Thr20 and Ala33 in hHSF1 are evolutionarily conserved in mouse HSF1, but not in HSF1 orthologs of invertebrate species (Supplementary Fig. 1e).

We next performed GST pull-down assay using hPARP13 deletion mutants and identified two HSF1-binding regions including a zinc finger domain (amino acids 77–110) and WWE domain (amino acids 605–689) in vitro (Supplementary Fig. 1f). We overexpressed HA-hPARP13 mutants, which lacked one of the two HSF1-binding regions (ΔZ and ΔWWE) or both regions (ΔZ–ΔWWE) (Supplementary Fig. 1g) in HEK293 cells,

**Fig. 1** HSF1, PARP13, and PARP1 form a ternary complex. **a** HEK293 cells were transfected with HA-tagged PARP proteins, and subjected to HSF1 immunoprecipitation and immunoblotting. Asterisks indicate the full-length products from the transfected PARP constructs. **b** hPARP13-His purified from bacteria was pulled down with purified GST, GST-hPARP1, GST-hHSF1, GST-hHSF2, or GST-hHSF4, and subjected to immunoblotting. Input of PARP13-His was also shown at the bottom. Asterisks indicate full-length GST-fusion proteins. **c** Endogenous immunoprecipitation in nuclear fractions of HeLa cells, in which PARP1 or PARP13 was knocked down by infection with adenovirus expressing the corresponding shRNA, or scrambled RNA (SCR) as control. Input: PARP1, 1%; PARP13, 5%; HSF1, 1%. Immunoprecipitate (IP), 100%. Percentages of co-precipitated PARPs are shown. Nuclear (N) and cytoplasmic (C) fractions were blotted using an antibody for nuclear (SP1) or cytoplasmic (HSP90) protein. **d** The DNA-binding domain of HSF1 interacts with PARP13. GST-pull-down from mixtures of purified hPARP13-His and GST-fused hHSF1 mutants was performed, and proteins were subjected to immunoblotting. DBD DNA-binding domain; HR hydrophobic heptad repeat; DHR downstream of HR-C. **e** Alignment of the amino-acid sequences of the DNA-binding domains in hHSF1, hHSF2, and hHSF4. Thirteen residues in HSF1 differing among three sequences were substituted with alanine (black dots) or glutamic acid (white dots). Predicted secondary structure including four α-helices (boxes H1 to H3 and a C-terminal α-helix C-H), four β-sheets (arrows β1 to β4), and a wing motif is shown. **f** HEK293 cells were transfected with wild-type and mutated hHSF1-HA, in which Thr20 or Ala33 was replaced with a corresponding amino acid from hHSF2 or hHSF4. Extracts of these cells were subjected to HA immunoprecipitation and immunoblotting. **g** HEK293 cells were transfected with wild-type and mutated HA-hPARP13 (ΔZ, ΔWWE, and ΔZ-ΔWWE). Extracts of these cells were subjected to HA immunoprecipitation and immunoblotting

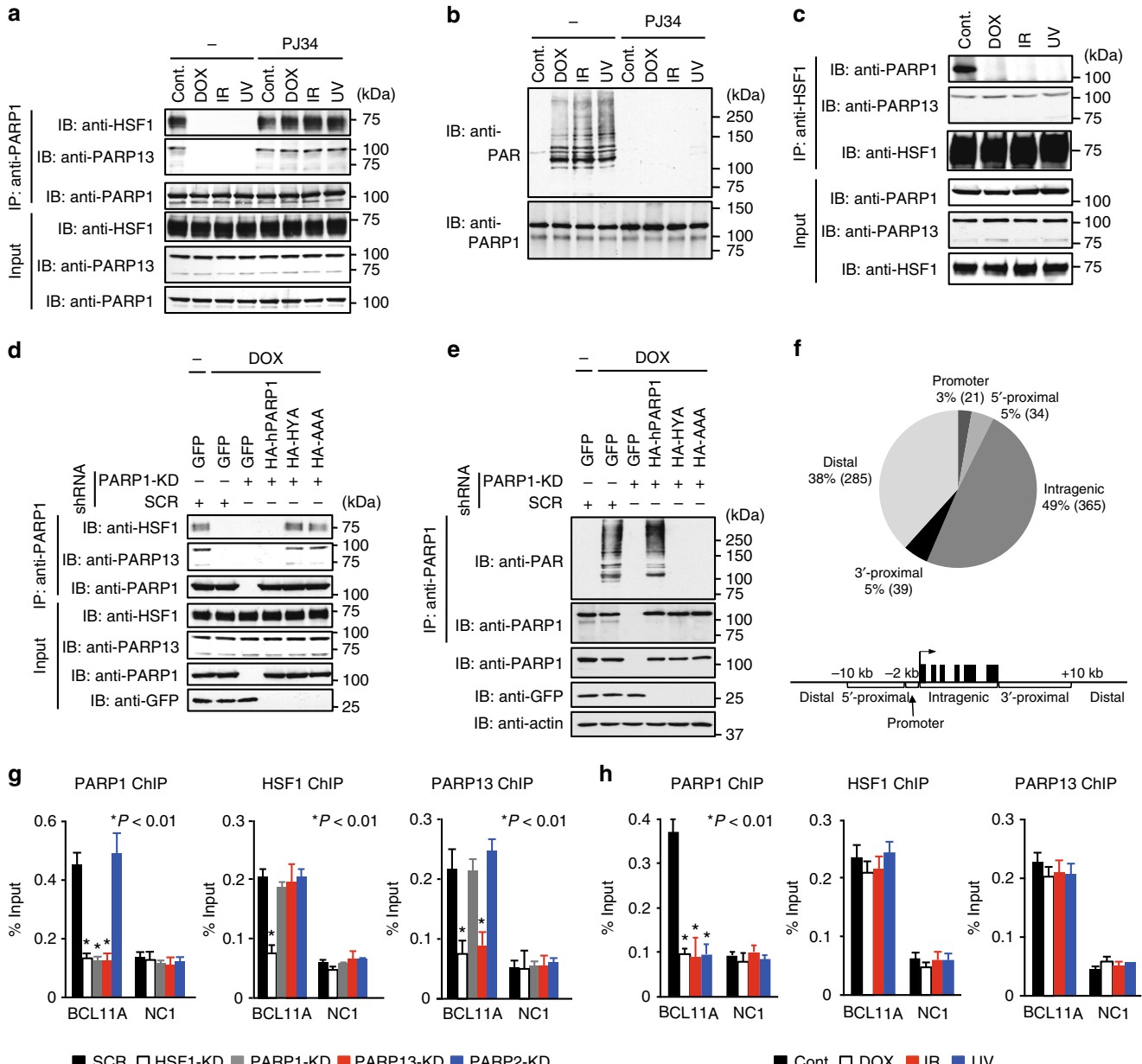

**Fig. 2** PARP1 activity regulates HSF1–PARP13–PARP1 ternary complex formation. **a** Extracts of cells treated with DOX, IR, or UV in the presence or absence of PJ34 were subjected to PARP1 immunoprecipitation, and then to immunoblotting. **b** Denatured extracts of cells treated as described in **a** were subjected to PARP1 immunoprecipitation, and then to immunoblotting using PAR antibody. **c** Extracts of cells treated as described in **a** were subjected to HSF1 immunoprecipitation, and then to immunoblotting. **d** Cells, in which endogenous PARP1 was replaced with wild-type or its inactive mutant, were treated with DOX. Cell extracts were subjected to PARP1 immunoprecipitation and immunoblotting. **e** Denatured extracts of cells treated as described in **d** were subjected to PARP1 immunoprecipitation, and then to immunoblotting using PAR antibody. **f** Genomic distribution of PARP1. ChIP-seq analysis was performed using LMB-treated cells overexpressing HA-hPARP1 and hHSF1-HA, and a total of 744 PARP1 peaks were identified. Percentages and numbers of PARP1 peaks at genomic regions relative to nearby genes are shown. Among 744 PARP1 peaks, only 10 peaks (Intragenic, 4; 3′-Proximal, 1; Distal, 5) were identified after PARP13 knockdown. **g** ChIP assay of PARP1, PARP13, and HSF1 at *BCL11A* locus in HeLa cells, in which PARP1, PARP13, HSF1, or PARP2 were knocked down. ChIP-qPCR on the peak region (BCL11A) and negative control region (NC1) was performed ($n = 3$). **h** ChIP assay at *BCL11A* locus in HeLa cells treated with DOX, IR, or UV ($n = 3$). Mean ± s.d. is shown. Asterisks indicate $P < 0.01$ by Student's *t*-test

and performed a co-precipitation experiment. HSF1 was not co-precipitated with any of the three HA-hPARP13 mutants, which suggests that both the zinc finger and WWE domains in PARP13 are required for stable interaction with HSF1 in vivo (Fig. 1g). PARP1 was co-precipitated with all the PARP13 mutants, which confirms that the zinc finger and WWE domains are not required for PARP1-PARP13 interaction (Fig. 1g).

**PARP1 activity regulates HSF1–PARP13–PARP1 complex formation**. PARP1, but not PARP13, possesses a C-terminal catalytic domain that synthesizes PAR on target proteins including PARP1 itself[3, 34]. Because auto-PARylation of PARP1 regulates the interaction with proteins involved in chromatin remodeling and DNA repair, we examined whether auto-PARylation regulates formation of the HSF1–PARP13–PARP1

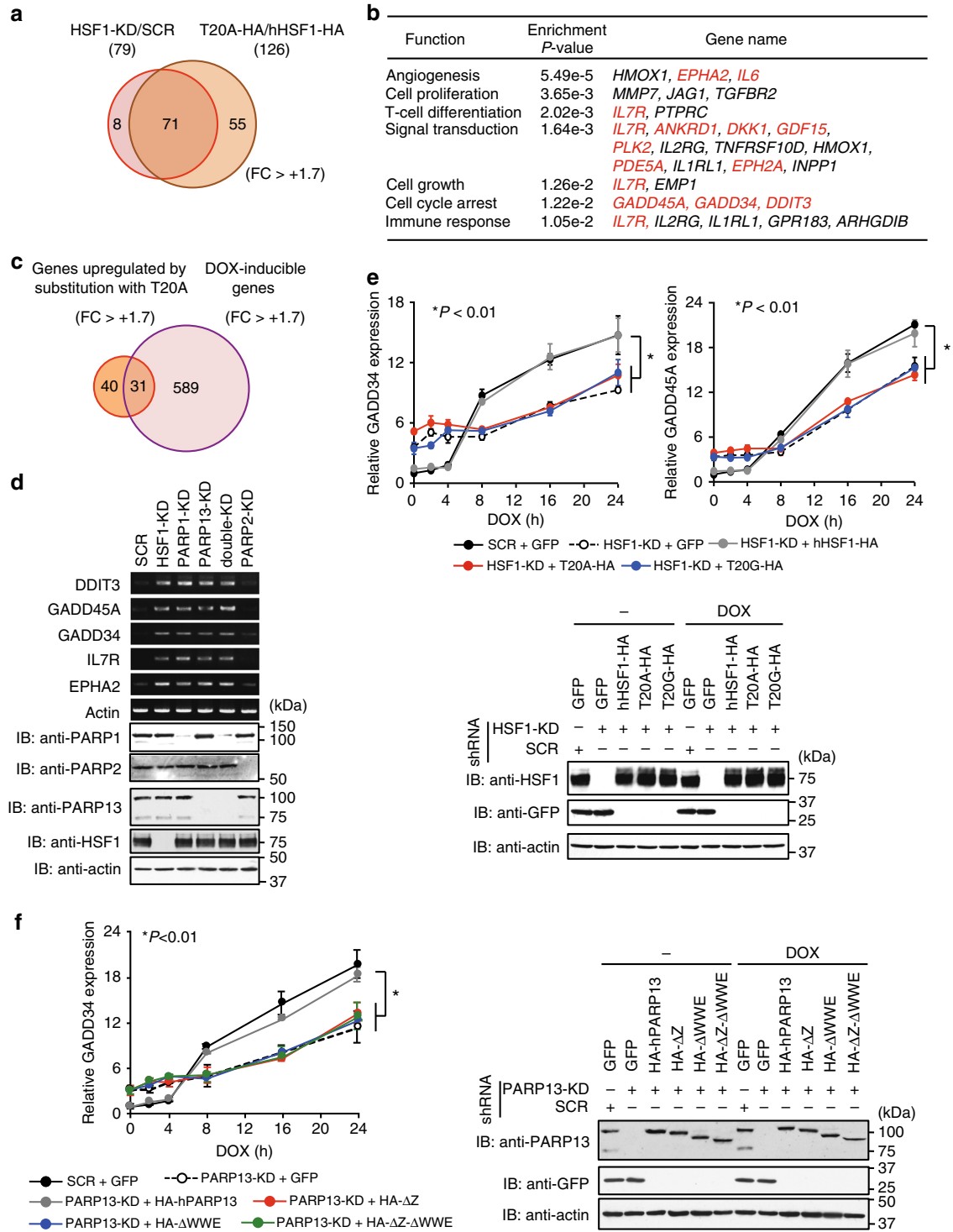

**Fig. 3** HSF1–PARP13–PARP1 enhances DNA damage-induced gene expression. **a** Venn diagram showing genes upregulated by HSF1 knockdown and substitution with hHSF1-T20A in HeLa cells (fold change > +1.7; P < 0.05; n = 3). **b** Gene ontology enrichment analysis of 71 genes upregulated by substitution with hHSF1-T20A. The P-values and gene names associated with each functional category are shown. Names of DNA-damage-inducible genes are indicated in red. **c** Venn diagram showing genes upregulated by DOX treatment and by substitution with hHSF1-T20A (fold change > +1.7; P < 0.05; n = 3). **d** Expression of DNA-damage-inducible genes in unstressed condition. RT-PCR analysis of five genes was performed using cells knocked down for HSF1, PARP1, PARP13, or both PARP1 and PARP13 (double-KD) (upper). Cell extracts were also subjected to immunoblotting (lower). **e** Expression of GADD genes in cells expressing HSF1 mutants. Cells, in which endogenous HSF1 was replaced with each protein, were treated with DOX for the indicated periods. GADD mRNA levels were quantified by RT-qPCR (n = 3) (upper). Mean ± s.d. is shown. Asterisks indicate P < 0.01 by ANOVA. Extracts from cells before and after treatment were subjected to immunoblotting (lower). **f** Expression of GADD34 gene in cells expressing PARP13 mutants. Cells, in which endogenous PARP13 was replaced with each mutant, were treated with DOX for the indicated periods. GADD34 mRNA levels were quantified by RT-qPCR (n = 3) (left). Mean ± s.d. is shown. Asterisks indicate P < 0.01 by ANOVA. Extracts from cells before and after the treatment were subjected to immunoblotting (right)

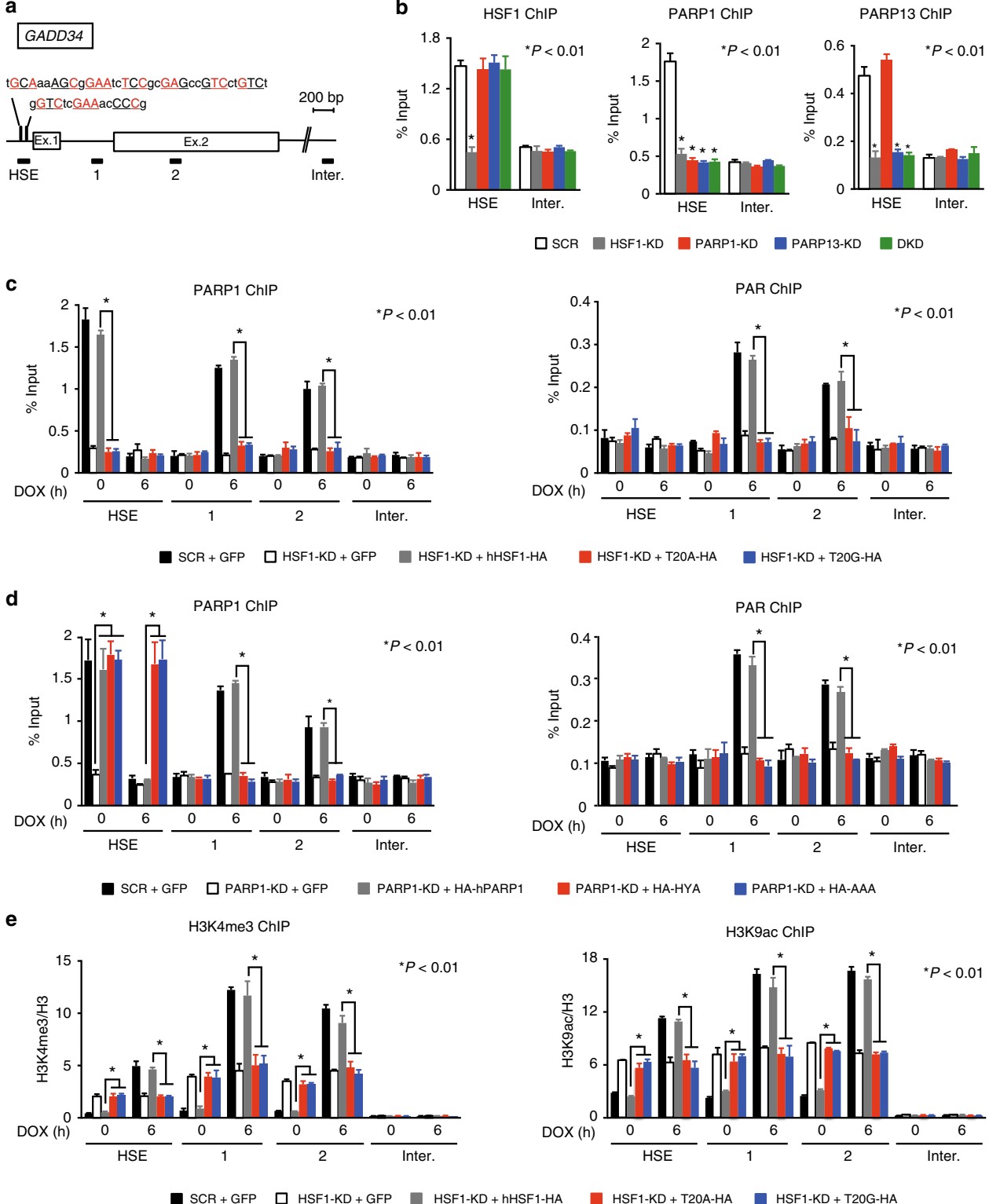

**Fig. 4** HSF1–PARP13 promotes PARP1 redistribution and chromatin opening. **a** Schematic view of HSEs in the promoter of the *GADD34* gene. Bars indicate amplified DNA regions by ChIP-qPCR. Exons and HSE sequences are shown. Consensus nGAAn units are underlined and conserved sequences are highlighted in red. **b** Constitutive occupancy of HSF1, PARP1, and PARP13 on the HSEs. ChIP-qPCR was performed after each knockdown ($n = 3$). Double knockdown (DKD) indicates knockdown of both PARP1 and PARP13. **c** Occupancy of PARP1 and PAR on the *GADD34* locus in cells expressing each hHSF1 mutant. ChIP-qPCR on the HSE and each gene locus (region 1 or 2) was performed before and after DOX treatment ($n = 3$). **d** Occupancy of PARP1 and PAR on the *GADD34* locus in cells expressing each hPARP1 mutant. ChIP-qPCR was performed before and after DOX treatment ($n = 3$). **e** Levels of active chromatin marks on the *GADD34* locus in cells expressing each hHSF1 mutant. ChIP-qPCR was performed before and after DOX treatment ($n = 3$). Mean ± s.d. is shown. Asterisks indicate $P < 0.01$ by Student's $t$-test

ternary complex. Treatment of cells with DNA damage reagents, such as doxorubicin (DOX), IR, or UV, induced auto-PARylation of PARP1 and its dissociation from the HSF1–PARP13 complex (Fig. 2a, b, first four lanes), which was stable during DNA damage (Fig. 2c). DNA damage-induced PARP1 dissociation was blocked by treatment with the PARylation inhibitor PJ34 (Fig. 2a, b, last four lanes), or replacement with hPARP1 mutants lacking PARylation activity (HYA, AAA) (Fig. 2d, e and Supplementary Fig. 2a). Dissociation of the HSF1–PARP13–PARP1 ternary complex by replacement with hHSF1 mutants or knockdown of PARP13 did not induce auto-PARylation of PARP1 (Supplementary Fig. 2b). These results indicate that auto-PARylation of PARP1 regulates its dissociation from HSF1–PARP13 during DNA damage.

To investigate the possibility that HSF1–PARP13 recruits PARP1 to the genome, we performed PARP1 chromatin immunoprecipitation sequencing (ChIP-seq) analysis using LMB-treated HeLa cells overexpressing hPARP1 and hHSF1 (Supplementary Fig. 2c). A total of 744 PARP1-binding peaks were identified; nearly 60% of peaks were found within promoters and bodies of annotated genes, and 38% of peaks were found in distal regions (Fig. 2f)[37]. Only 10 peaks were identified in these cells after PARP13 knockdown. ChIP assays confirmed binding of endogenous PARP1 at four arbitrarily chosen sites including the promoter of the *BCL11A* gene, and showed that HSF1 and PARP13 bound to the same sites (Fig. 2g and Supplementary Fig. 2d–f). Knockdown of HSF1 or PARP13, but not PARP2, abolished PARP1 binding at these sites. Furthermore, treatment with DOX, IR, or UV reduced PARP1 binding to the *BCL11A* promoter (Fig. 2h). These results suggest that HSF1–PARP13 recruits PARP1 to genomic regions including gene promoters.

**HSF1–PARP13–PARP1 enhances gene expression upon DNA damage**. To better understand whether the HSF1–PARP13–PARP1 ternary complex regulates gene expression, we performed DNA microarray analysis using HeLa cells, in which HSF1 was knocked down or substituted with wild-type hHSF1 or hHSF1-T20A (Supplementary Fig. 3a). Although HSF1 is generally thought to act as an activator, HSF1 knockdown not only reduced expression of many genes but also increased the expression of a substantial number of other genes (Supplementary Fig. 3b). Among 79 upregulated genes in HSF1-knockdown cells, the expression of 71 genes (90%) was also elevated by substitution with hHSF1-T20A (Fig. 3a and Supplementary Fig. 3c). Constitutive expression of heat shock genes including *HSP70* and *HSP40* was not altered by this substitution (Supplementary Fig. 3d). Gene ontology enrichment analysis for the upregulated genes included many DNA damage-inducible genes, such as *GADD45A*, *GADD34*, *DDIT3*, *IL17R*, and *EPHA2* (Fig. 3b), whose products are not DNA repair factors but rather involved in the regulation of the cell cycle, apoptosis, stress signaling, and protein synthesis during DNA damage[38–40]. The expression of about half of these genes (31 genes) was induced by DOX treatment (Fig. 3c). We confirmed that the expression levels of these genes were increased by knockdown of HSF1, PARP1, or PARP13, but not by knockdown of PARP2 (Fig. 3d). These results indicate that the HSF1–PARP13–PARP1 ternary complex suppresses constitutive expression of a subset of DNA damage-inducible genes.

We next analyzed the profiles of DNA damage-induced expression of *GADD34* and *GADD45A*. In HSF1 knockdown cells or cells expressing hHSF1 mutants, induced expression of *GADD34* and *GAGG45A* was markedly suppressed at 16 and 24 h after DOX treatment, whereas basal expression levels were elevated (Fig. 3e). Similar expression profiles of *GADD34* were observed in PARP13 knockdown cells or cells expressing HA-

hPARP13ΔZ, HA-hPARP13ΔWWE, or HA-hPARP13ΔZ-ΔWWE, which cannot interact with HSF1 (Fig. 3f). In cells expressing hPARP1 mutants (HYA, AAA), basal levels of *GADD34* expression were unaltered, but induced expression was markedly suppressed (Supplementary Fig. 3e). PARP1 knockdown elevated the basal level of *GADD34* expression like with knockdown of HSF1 or PARP13. Furthermore, we generated HeLa cells harboring a luciferase reporter driven by the human *GADD34* promoter (Fig. 6h)[41], found that basal luciferase activity was elevated, and that DOX-induced activity was reduced by the substitution of endogenous HSF1 with hHSF1 mutants (Supplementary Fig. 3f). These results demonstrate that HSF1–PARP13–PARP1 ternary complex suppresses constitutive expression of *GADD34* and enhances its induction during DNA damage, and suggest that PARP1 mediates repressive activity of HSF1 under unstressed conditions.

**HSF1–PARP13 promotes PARP1 redistribution to active genes**. To address the question of how the HSF1–PARP13–PARP1 ternary complex promotes DNA damage-induced gene expression, we examined the occupancy of each component at the *GADD34* locus. We identified HSE sequences in the promoter that were bound by the ternary complex in a manner dependent on HSF1 (Fig. 4a, b), and showed that only PARP1 occupancy was lost after DOX treatment (Supplementary Fig. 4a). We hypothesized that PARP1 may redistribute to the gene locus after DNA damage, because *Drosophila* PARP redistributes from the 5′ end of the *HSP70* gene to throughout the *HSP70* locus during heat shock[13]. We found that PARP1 redistributed from the HSE to regions 1, 2, and 3 on the *GADD34* locus at 6 h after DOX treatment, and then disappeared at 12 h (Supplementary Fig. 4b). The transient redistribution of PARP1 on this locus was accompanied by PARylation of chromatin (Supplementary Fig. 4b). PARP13 knockdown abolished not only PARP1 occupancy on the HSE in unstressed conditions, but also its redistribution to the gene locus and PARylation of the locus (Supplementary Fig. 4c). Constitutive occupancy of PARP1 and DOX-induced redistribution of PARP1 and chromatin PARylation were also abolished in HSF1 knockdown cells, or cells expressing hHSF1-T20A or hHSF1-T20G (Fig. 4c). When endogenous PARP1 was substituted with HA-hPARP1-HYA and HA-hPARP1-AAA, they remained binding to the HSE and did not redistributed across the *GADD34* locus during DOX treatment (Fig. 4d). HSF1 occupied the HSE before and after DOX treatment and did not redistribute on the *GADD34* locus (Supplementary Fig. 4d). We next investigated the chromatin status in cells expressing hHSF1 mutants and found that active chromatin marks such as H3K4me3 and H3K9ac were elevated in unstressed conditions, but were reduced after DOX treatment (Fig. 4e and Supplementary Fig. 4e). These results indicate that the ternary complex is required for PARP1 redistribution into *GADD34* locus during DOX treatment, and promotes the establishment of an active chromatin state.

We examined whether two zinc finger domains (Zn1 and Zn2) of PARP1, which recognize DNA breaks[42], is required for redistribution into the *GADD34* locus or not. Substitution of endogenous PARP1 with HA-hPARP1ΔZ1 (deletion of Zn1) or HA-hPARP1ΔZ2 did not affect constitutive PARP1 occupancy in the *GADD34* promoter, PARP1 redistribution into the *GADD34* locus during DOX treatment, and chromatin PARylation (Supplementary Fig. 4f). HA-hPARP1ΔZ1-2, which lacked both Zn1 and Zn2, bound to the *GADD34* promoter at a lower level in unstressed condition, but still redistributed into the gene locus during DOX treatment. These results suggest that PARP1 redistribution into the *GADD34* locus is not related with DNA damage recognition[43].

**HDAC1 maintains HSF1–PARP13–PARP1 complex on gene promoters**. HSF1–PARP13–PARP1 ternary complex formation is regulated by PARP1 activity, which is modulated by post-translational modifications including acetylation[3], and PARP1 and HSF1 can interact with the histone deacetylase HDAC1[44–46]. To uncover the mechanism that maintains the ternary complex constitutively on gene promoters, we examined the role of

HDAC1-mediated deacetylation of PARP1. We found that HDAC1 was a component of the HSF1–PARP13–PARP1 complex in untreated cells, but not in DOX-treated cells (Fig. 5a). HDAC1 occupied the HSE on the *GADD34* promoter, and its level decreased during DOX treatment (Fig. 5b). We examined acetylation of PARP1 in unstressed conditions and found that PARP1 was constitutively acetylated at a high level in cells

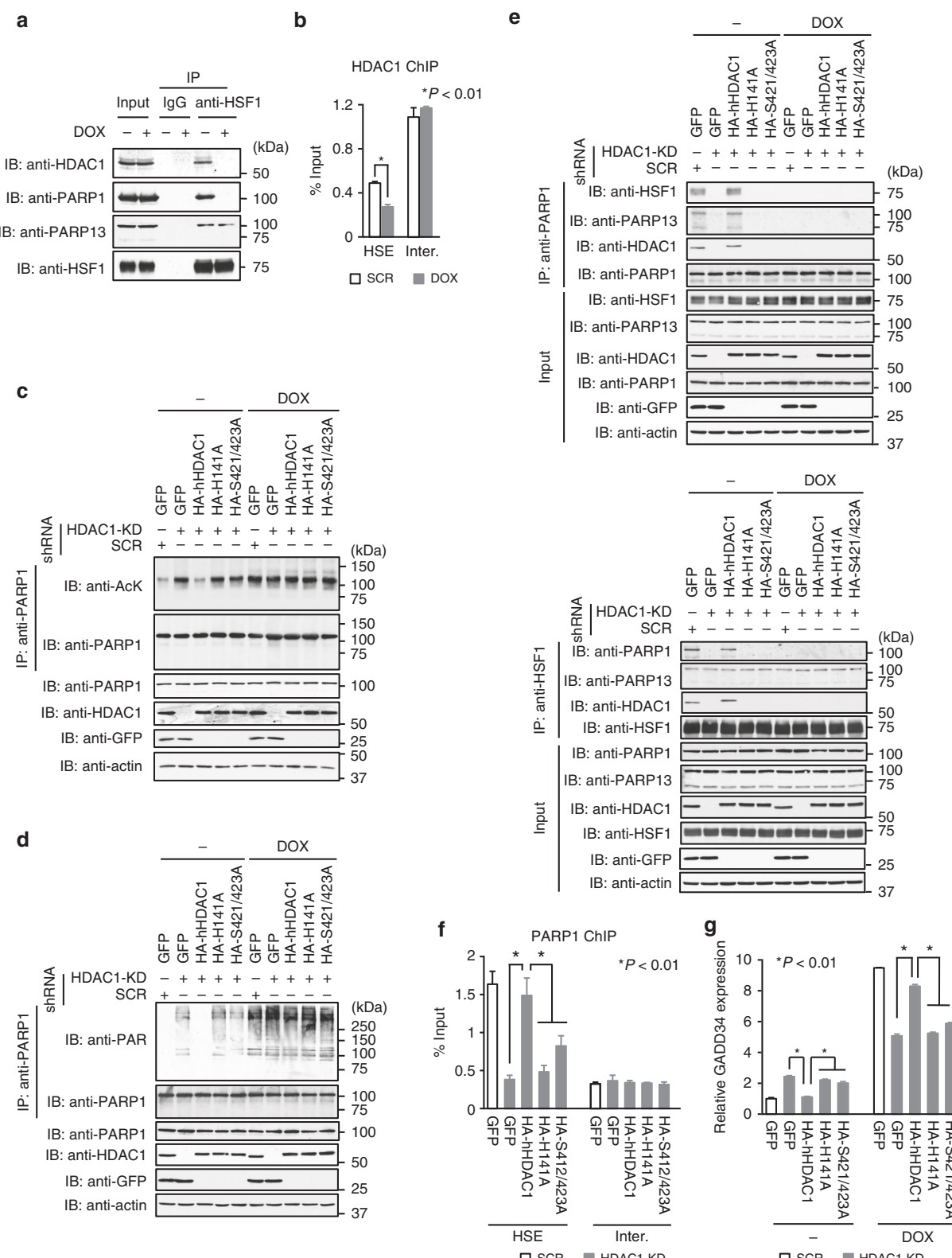

deficient in HDAC1 or cells expressing hHDAC1 mutants with reduced deacetylase activity (H141A, S421/423A) (Fig. 5c). In the same cells, PARP1 was constitutively auto-PARylated (Fig. 5d) and HDAC1 was released from PARP1 (Fig. 5e). Importantly, the release of HDAC1 was associated with dissociation of PARP1 from HSF1–PARP13 (Fig. 5e) and from the HSE on the *GADD34* promoter in unstressed conditions (Fig. 5f), and was accompanied by elevated expression of *GADD34* (Fig. 5g). These results suggest that HDAC1 maintains ternary complex occupancy on gene promoters by deacetylating PARP1.

During DOX treatment, PARP1 was acetylated and auto-PARylated (Fig. 5c, d) and HDAC1 was released from PARP1 (Fig. 5e). HDAC1 was released even from inactive hPARP1 mutants, suggesting a mechanism that was independent of auto-PARylation (Supplementary Fig. 5a, b). Expression of *GADD34* during DOX treatment was also reduced in cells expressing hHDAC1 mutants (Fig. 5g), like in cells expressing hHSF1-T20A or hHSF1-T20G (Fig. 3e).

**HSF1–PARP13–PARP1 promotes DNA repair**. In addition to activating DNA damage-inducible genes, cells cope with DNA damage by repairing DNA lesions, where signaling and repair proteins assemble in a sequential and coordinated manner. Among these, PARP1 is one of the first proteins recruited to DNA breaks including DSBs, which are repaired by HRR and NHEJ[4, 6]. Furthermore, HSF1 deficiency results in impaired DNA repair[47]. To investigate whether the HSF1–PARP13–PARP1 ternary complex is involved in the repair of DNA damage, we examined accumulation of repair factors into foci by immunofluorescence. We found that γH2AX foci appeared at 6 h after DOX treatment (Supplementary Fig. 6a), and the signal intensity of γH2AX and number of RAD51 and 53BP1 foci after DOX treatment were reduced in cells lacking HSF1 or cells expressing hHSF1 mutants (Fig. 6a–c and Supplementary Fig. 6b–d)[48, 49]. The percentages of cells in S and G2/M phases after DOX treatment were similar among these cells (Supplementary Fig. 6e). Production of reactive oxygen species and protein levels of DNA damage-response kinases such as ATM, ATR, and DNA-PK during DOX treatment were similar (Supplementary Fig. 6f, g). The signal intensity of γH2AX and number of RAD51 and 53BP1 foci after DOX treatment were also reduced in PARP13 knockdown cells (Supplementary Fig. 6h–j). A neutral comet assay was conducted to estimate overall DNA damage. In control conditions, both scramble RNA-treated and HSF1-knockdown cells had no tail intensity, which indicates little or no DNA damage (Fig. 6d). During DOX treatment, scramble RNA-treated cells had low tail moment values, whereas HSF1-knockdown cells had high tail moment values (Fig. 6d). Re-expression of hHSF1, but not hHSF1 mutants, reduced the elevated tail moment values in HSF1-knockdown cells. Thus, loss of HSF1 or PARP13 results in reduced levels of γH2AX signal, reduced recruitment of 53BP1 and RAD51, and impairment of DNA repair.

**HSF1–PARP13 facilitates PARP1 redistribution to DNA lesions**. To further estimate PARP1 accumulation at DNA damage sites, we used cells with a single copy of a pDR-GFP reporter, in which HRR of an I-SceI-induced DSB-activated GFP expression (Supplementary Fig. 7a)[50, 51]. PARP1 accumulated at an SCE-1 region near the I-SceI cutting site, but not at other upstream regions (Supplementary Fig. 7b, c). We found that accumulation of PARP1 and γH2AX in the SCE-1 region during HRR is markedly reduced in cells lacking HSF1 or cells expressing hHSF1-T20A (Fig. 6e, f). Furthermore, impaired accumulation of PARP1 was associated with reduced efficiency of HRR (Fig. 6g). Similar effects of impaired ternary complex formation on NHEJ were observed using cells with a pEJSSA reporter (Supplementary Fig. 7d–f)[52, 53]. These results demonstrate that HSF1–PARP13 facilitates redistribution of PARP1 close to an engineered DSB site and improves DNA repair efficiency, probably through PAR-mediated signaling and downstream recruitment of DNA repair factors.

Because we hypothesized that HSF1 is binding to pDR-GFP and pEJSSA reporters, we decided to test a pGADD34-Luc reporter, which has an I-SceI cleavage site, and an HSE-deleted pGADD34ΔHSE-Luc reporter (Fig. 6h). HSF1 and PARP1 bound to the HSEs on both the endogenous *GADD34* promoter and the reporter in HeLa-pGADD34-Luc cells (Fig. 6i). After the expression of I-SceI, PARP1 binding at the HSEs, but not HSF1 binding, was reduced to a level similar to that in HeLa-pGADD34ΔHSE-Luc cells. Simultaneously, PARP1 markedly accumulated at the LUC region of the reporter (2.26-fold) in HeLa-pGADD34-Luc cells, but accumulated only a little (1.55 fold) in HeLa-pGADD34ΔHSE-Luc cells. These results suggest that PARP1 bound to the HSE in the pGADD34-Luc reporter redistributed to adjacent DNA damage sites.

**HSF1–PARP13–PARP1 protects cells from genotoxic stress**. HSF1 is required for human and mouse tumor cell proliferation[25, 32], and silencing of HSF1 enhances sensitivity to DNA damage[47]. To investigate the impact of the HSF1–PARP13–PARP1 ternary complex, we substituted endogenous HSF1 with hHSF1-T20A or hHSF1-T20G in HeLa cells and monitored survival of these cells during DOX treatment. Substitution with hHSF1 mutants reduced cell survival during DOX treatment (Supplementary Fig. 8a), but did not affect normal cell growth (Fig. 7a). Clonogenic survival after DOX treatment was significantly reduced by substitution with hHSF1 mutants (Supplementary Fig. 8b). Thus, the ternary complex protects tumor cells from exposure to DOX.

Interestingly, PARP1 activity is indispensable for normal proliferation of mammary tumor cells, in which a gene encoding an HRR factor BRCA1 is mutated, because PARP1 inhibition blocks alternative DNA repair pathways including NHEJ[5]. We hypothesized that the ternary complex may play a role in BRCA1-deficient tumor cell growth. Substitution with hHSF1 mutants had little effect on proliferation of human HeLa cells and MCF7 mammary tumor cells (Fig. 7a and Supplementary Fig. 8c). In contrast, this substitution moderately reduced proliferation of human mammary tumor HCC1937 and MDA-MB-436 cells,

**Fig. 5** HDAC1 maintains the HSF1–PARP13–PARP1 complex on gene promoters. **a** Extracts of HeLa cells untreated (−) or treated (+) with DOX were subjected to HSF1 immunoprecipitation and immunoblotting. **b** HDAC1 occupancy on the HSE of the *GADD34* locus. ChIP-qPCR was performed before and after DOX treatment ($n = 3$). Mean ± s.d. is shown. Asterisks indicate $p < 0.01$ by Student's *t*-test. **c, d** Endogenous HDAC1 was substituted with GFP, HA-hHDAC1, HA-hHDAC1-H141A, or HA-hHDAC1-S421/423A. Denatured extracts of cells, untreated or treated with DOX, were subjected to PARP1 immunoprecipitation and immunoblotting using antibodies including anti-acetyl lysine antibody (anti-AcK) (**c**) and anti-PAR antibody (**d**). **e** Extracts of untreated or DOX-treated cells, in which endogenous HDAC1 was substituted with a series of mutants, were subjected to immunoprecipitation of PARP1 (upper) or HSF1 (lower) and to immunoblotting. **f** PARP1 occupancy on HSE in cells expressing hHDAC1 mutants. ChIP-qPCR was performed ($n = 3$). Mean ± s.d. is shown. Asterisks indicate $P < 0.01$ by Student's *t*-test. **g** Expression of *GADD34* in cells expressing HDAC1 mutants. Cells, in which endogenous HDAC1 was replaced, were treated or not with DOX for 16 h. GADD34 mRNA level was quantified by RT-qPCR ($n = 3$). Mean ± s.d. is shown. Asterisks indicate $P < 0.01$ by Student's *t*-test

which carry *BRCA1* mutations[54]. In these tumor cells, HSF1 and PARP1 formed a complex in a manner dependent on PARP13 (Supplementary Fig. 8d). Furthermore, the substitution reduced proliferation of *BRCA1*$^{-/-}$*p53*$^{-/-}$ mouse mammary tumor KB1P-G3 and KB1P-B11 cells[55] to the same level as HSF1 knockdown (Fig. 7b and Supplementary Fig. 8e). To test whether impaired

ternary complex formation promotes DNA damage in normal growth conditions, we used comet assays. We found that KB1P-G3 cells showed a weak but distinct tail intensity even in unstressed conditions, and the number of these DNA-damaged cells increased by HSF1 knockdown or substitution with hHSF1 mutants (Fig. 7c). We further performed isograft experiments in

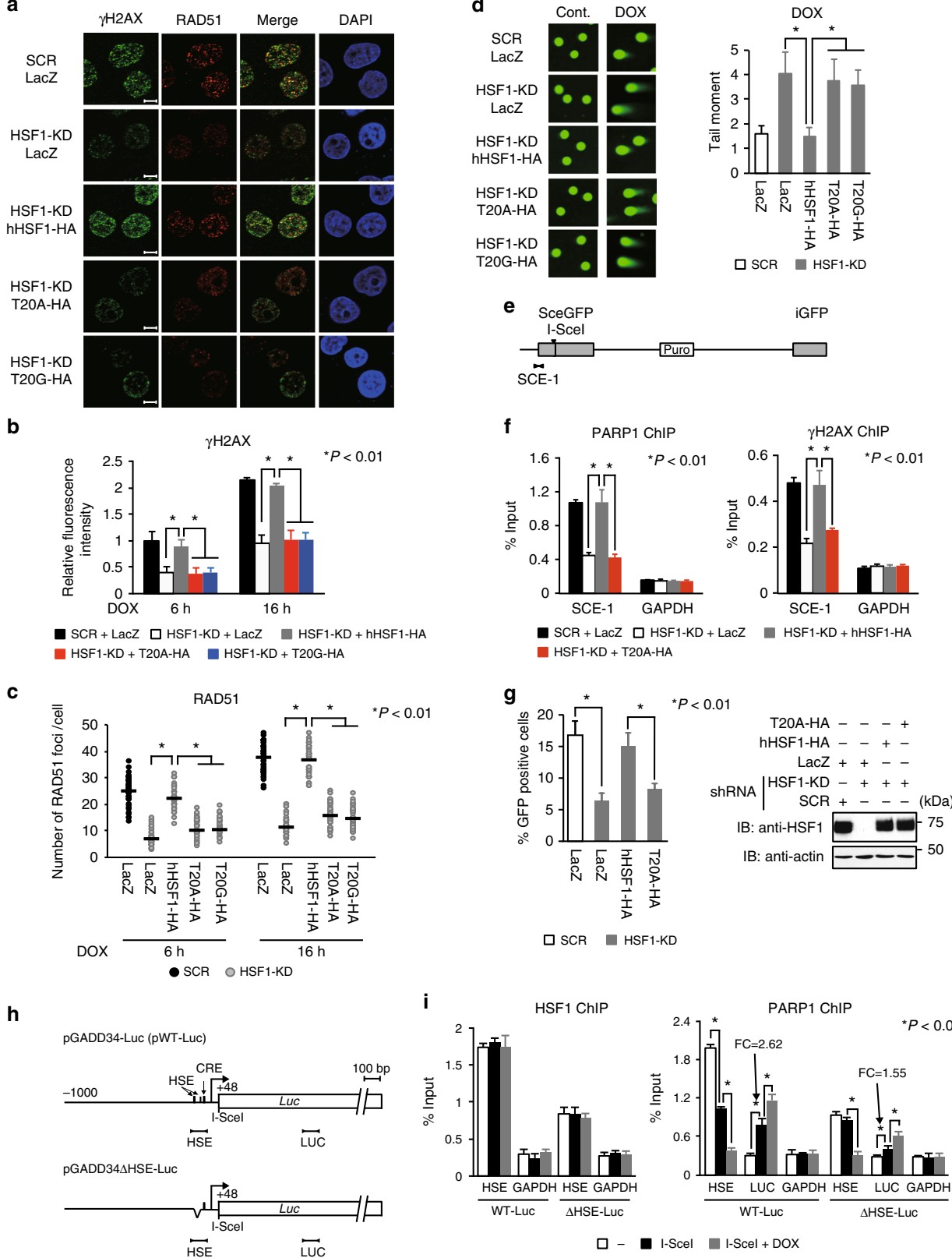

which KB1P-G3 cells were injected into FBV/N mice. Tumor formation of KB1P-G3 cells harboring one of the hHSF1 mutants was significantly reduced compared with those harboring wild-type hHSF1 (Fig. 7d). These results indicate that the ternary complex specifically supports growth of BRCA1-deficient mammary tumors partly by facilitating DNA repair.

## Discussion
The HSR is a primitive adaptive response to proteotoxic stresses including high temperature, and is mainly regulated by the universally conserved HSF1 in mammals[24, 56]. The primary function of HSF1 is to maintain the proteome balance in a cell by inducing HSPs and non-HSP proteins involved in protein degradation, and loss of HSF1 activity is closely associated with the progression of aging and age-related neurodegenerative diseases[17]. In this study, we identified HSF1 as a pivotal regulator of genome integrity in cooperation with PARP1, which plays fundamental roles during DDR (Fig. 7e).

PARP1 is a multifunctional regulator of chromatin structure, transcription, and DNA repair, and consists of multiple domains including an N-terminal DNA-binding domain and a C-terminal catalytic domain[3, 5]. It is unevenly distributed in the chromatin of normally growing cells[37, 57], and redistributes to DNA lesions during DNA damage[4, 6, 7] and to inducible gene loci in response to stimuli such as heat shock and hormone treatment in a manner dependent on PARP1 activation[12–14]. However, mechanisms of constitutive binding to chromatin and stress-induced PARP1 redistribution are not fully understood in mammals. Here we identified the HSF1–PARP13–PARP1 ternary complex in a set of PARP1 peak sites identified by ChIP-seq (Fig. 2). PARP1 binding at these sites is dependent on HSF1 and PARP13, suggesting that HSF1–PARP13 recruits PARP1 to HSF1-binding regions, which are widely distributed in the genome[27–30]. Although PARP1 may not be able to access certain chromatin states via its DNA-binding domain (made up of zinc finger domains)[43], HSF1–PARP13 could tether PARP1 to these areas to remodel chromatin. Remarkably, HSF1 deficiency or impaired ternary complex formation abolishes constitutive PARP1 binding to these sites (Fig. 2), and markedly reduces the redistribution of PARP1 to DNA damage sites and the accumulation of DNA repair factors including RAD51 and 53BP1 (Fig. 6). Furthermore, it reduces DNA repair efficiency. On the other hand, we also found that HSF1 promotes the induction of a set of DNA damage-inducible genes including *GADD34* (Fig. 3). PARP1 redistributes from the promoter of *GADD34* to its gene locus during DNA damage, and impaired ternary complex formation blocks the redistribution of PARP1, PARylation of the chromatin, and establishment of an active chromatin state (Fig. 4). We propose that HSF1-mediated pre-recruitment of PARP1 to DNA facilitates the redistribution of

PARP1 during DNA damage. It is worth noting that lack of the ternary complex does not completely block PARP1 redistribution to DNA lesions, probably because PARP1 directly binds to the nucleosome[58, 59] and detects DNA breaks[60].

PARP13, which lacks PARylation activity, plays a key role in the formation and dissociation of the ternary complex. It is an RNA-binding protein that regulates stability and translation of viral RNA and cellular mRNA in the cytoplasm[34]. We showed that PARP13 shuttles between the nucleus and cytoplasm (Supplementary Fig. 1), and interacts with HSF1 as well as PARP1 in the nucleus (Fig. 1). PARP13 in the nucleus acts as a scaffold protein that regulates the association of PARP1 with HSF1. PARP1 dissociates from PARP13 by DNA damage-induced auto-PARylation (Figs. 2 and 4). HDAC1 maintains the interaction by inactivating PARP1 through deacetylation, and DNA damage-induced dissociation of PARP1 from PARP13 is associated with elevated acetylation levels of PARP1 (Fig. 5). It is known that PARP1 is acetylated by the acetyltransferases p300/CBP and PCAF and deacetylated by a number of deacetylases including SIRT1 and HDAC1[3]. Furthermore, HDAC1 acetylation is rapidly induced under various stress conditions and increased acetylation of HDAC1 reduces its deacetylase activity[61]. Although PARP1 is activated by binding to damaged DNA, it is also activated by acetylation during stress in a manner that is independent of DNA damage[62]. Our observations suggest that it is first activated in the ternary complex through an acetylation-mediated mechanism and is then released from that. Activation and auto-PARylation of PARP1 result in its release from chromatin, but modestly modified PARP1 may retain its association with chromatin[3]. Detailed mechanism of PARP1 dissociation from HSF1–PARP13 and its redistribution during DNA damage will be uncovered in future.

HSF1 promotes tumor initiation and progression, and supports proliferation of malignant tumor cells[25]. These cells suffer from chronic proteotoxic stress, which enhances formation of protein aggregates and amyloids, and HSF1 supports cell proliferation by inhibiting aggregate formation and amyloidgenesis[26]. It is thought that HSF1 maintains the proteome balance by directly regulating the expression of genes that are involved in diverse biological process such as protein folding, protein translation, chromatin remodeling, and DNA repair[27]. On the other hand, malignant tumor cells need to adapt to some extent to continuous DNA damage in order to proliferate and the DDR plays a role in this process[1, 2]. BRCA1-null tumor cells, which are defective for HHR, require alternative pathways including NHEJ and base excision repair for SSBs to repair damaged DNA. These cells are highly sensitive to PARP inhibitors, because PARP1 is required for alternative repair pathways[1, 63]. We showed that impaired complex formation as well as HSF1 deficiency reduces proliferation of human mammary tumor cells carrying *BRCA1* mutations[54] and BRCA1-null mammary tumor cells[55], and

**Fig. 6** HSF1-mediated PARP1 redistribution promotes DNA repair. **a**–**c** HeLa cells, in which endogenous HSF1 was replaced with each mutant, were treated with 0.5 μM DOX, and then co-stained with γH2AX and RAD51 antibodies, and with DAPI. Fluorescence images, obtained using scanning confocal microscopy, and merged images (DOX for 16 h) are shown (**a**). Scale bar, 5 μm. Intensities of γH2AX fluorescence (**b**) and numbers of RAD51 foci (**c**) in 50 cells were estimated. Mean ± s.d. is shown. Asterisks indicate $P < 0.01$ by Student's *t*-test. **d** Cells treated as in **a** were exposed to DOX for 2 h, and then recovered for 2 h. DNA damage in these cells was measured using a neutral comet assay (25 cells), and tail moment values are shown. Mean ± s.d. is shown. Asterisks indicate $P < 0.01$ by Student's *t*-test. **e** Schematic structure of pDR-GFP reporter construct. SCE-1 region indicates an amplified region by ChIP-qPCR. **f** Accumulation of PARP1 and γH2AX in the SCE-1 region. HeLa-DRGFP cells, in which HSF1 was replaced as in **a**, were transfected with an I-SceI expression vector, and analyzed by ChIP assay ($n = 3$). Mean ± s.d. is shown. Asterisks indicate $P < 0.01$ by Student's *t*-test. **g** Efficiency of HR repair. Numbers of GFP-positive cells among 200 cells were counted, and the percentages of these cells are shown ($n = 3$) (left). Mean ± s.d. is shown. Asterisks indicate $P < 0.01$ by Student's *t*-test. HSF1 levels were examined by immunoblotting (right). **h** Schematic showing amplified regions, HSE and LUC, in the reporter constructs. The I-SceI cutting site is located upstream of the luciferase gene. The HSE and CRE elements are indicated. **i** Occupancy of HSF1 and PARP1 on the HSE and LUC regions during I-SceI treatment. The cells were transfected with pCBASce for 24 h, and ChIP-qPCR was performed. Some cells were treated with DOX for the last 8 h. Fold changes (FC) of PARP1 binding on the LUC region during I-SceI treatment are shown ($n = 3$). Mean ± s.d. is shown. Asterisks indicate $P < 0.01$ by Student's *t*-test

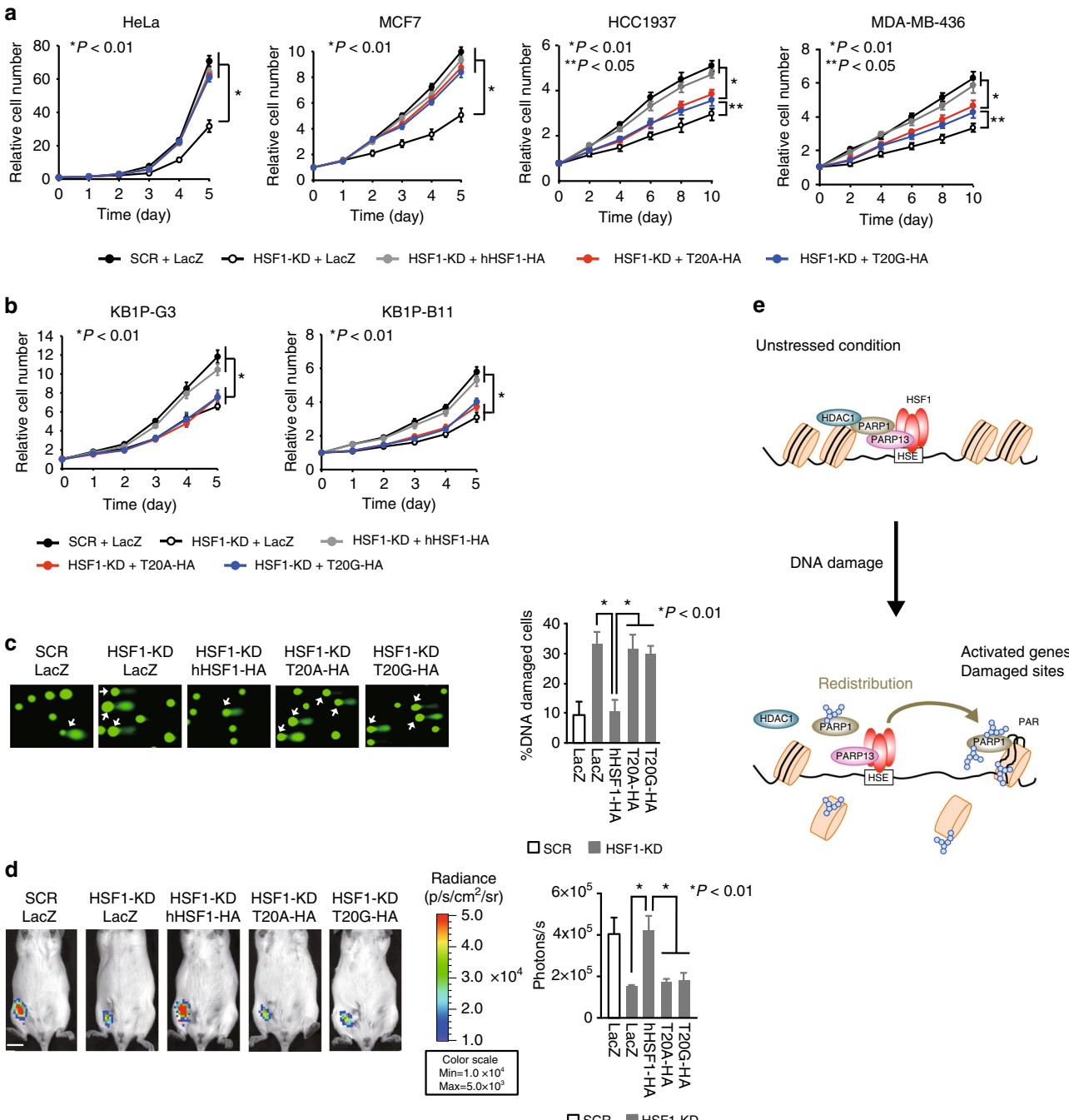

**Fig. 7** HSF1–PARP13–PARP1 protects cells from genotoxic stress. **a** Proliferation of human HeLa and three mammary tumor cells expressing hHSF1-T20A or hHSF1-T20G. Endogenous HSF1 was replaced with β-galactosidase (LacZ), hHSF1-HA, or its interaction mutants. Relative cell numbers at each time point are shown (n = 3). Mean ± s.d. is shown. Asterisks indicate P < 0.01 or 0.05 by ANOVA. **b** Proliferation of mouse KB1P-G3 and KB1P-B11 cells expressing hHSF1-T20A or hHSF1-T20G. Endogenous HSF1 was replaced as described in **a**, and relative cell numbers are shown (n = 3). **c** DNA damage in KB1P-G3 cells treated as in **b** was measured using a neutral comet assay. Percentages of cells with DNA damage showing DNA fluorescence in the tail (arrows) (100 cells) are shown (n = 3). Mean ± s.d. is shown. Asterisks indicate P < 0.01 by Student's t-test. **d** Tumor growth of KB1P-G3 cells. KB1P-G3-luc cells treated as in **b** were injected into the fourth left mammary fat pad of FBV/N mice, and tumor growth was evaluated by bioluminescence imaging after 4 weeks. Scale bar, 1 cm. Luciferase quantification is shown as photons per second (n = 5). Mean ± s.d. is shown. Asterisks indicate P < 0.01 by Student's t-test. **e** Schematic representation of the HSF1–PARP13–PARP1 ternary complex

enhances DNA damage in normal growth conditions (Fig. 7). Thus, HSF1-mediated DDR is a major mechanism of the addiction of BRCA1-null mammary tumors to HSF1, and could be a target for the treatment of a specific type of malignant tumors.

In this manuscript, we show that HSF1–PARP13–PARP1 ternary complex affects the DDR. The ternary complex may also modulate the HSR because PARP1 affects expression of *HSP70* during heat shock in *Drosophila* and mammalian cells[13, 14, 64]. It will be interesting to understand whether proteotoxic and genotoxic stresses mutually affect each other through the ternary complex, and how this complex contributes to the progression of age-related neurodegenerative diseases.

## Methods

**Plasmids and adenoviral vectors**. To generate an expression vector for each hemagglutinin (HA)-tagged human PARP protein, a cDNA fragment of PARP1 (flanked by *Not*I and *Xho*I sites), PARP2 (*Xho*I/*Hin*dIII), PARP5a (*Xho*I/*Hin*dIII), PARP7 (*Kpn*I/*Hin*dIII), or PARP12 (*Kpn*I/*Xho*I) was amplified by reverse transcription polymerase chain reaction (RT-PCR) using total RNA isolated from HeLa cells, and inserted into pShuttle-CMV vector (Stratagene). A cDNA fragment of PARP13 or PARP13S was amplified by RT-PCR and inserted into pcDNA3.1/Neo vector (Invitrogen) at a *Bam*HI/*Eco*RI site, and then inserted into pShuttle-CMV vector at a *Kpn*I/*Not*I site. Sequences of the pShuttle-HA-hPARPs were verified using 3500 Genetic Analyzer (Applied Biosystems). An expression vector pShuttle-hHSF1-HA and expression vectors for hHSF1-HA mutants and HA-hPARP13 were generated by PCR-mediated site-directed mutagenesis using mutated internal primers[32]. To generate expression vectors for HA-hPARP1-HYA and HA-hPARP1-AAA, in which three HYE amino acids in the catalytic domain were all substituted to HYA or AAA[65], a *Not*I/*Xho*I fragment of the mutated cDNA was inserted into the pShuttle-CMV (Stratagene) vector. Expression vectors for HA-hPARP1ΔZ1 (deletion of amino acid 6–91), HA-hPARP1ΔZ2 (deletion of amino acid 109–201), and HA-hPARP1ΔZ1-2, which lacked two zinc finger domains (Zn1 and Zn2) required for DNA break recognition[42], were similarly generated. pShuttle-HA-hHDAC1 was generated by inserting a *Hin*dIII/*Xho*I cDNA fragment into the pShuttle-CMV vector, and expression vectors for its mutants, HA-hHDAC1-H141A and HA-hHDAC1-S412/423A, which lacked catalytic activity and active phosphorylation, respectively[66, 67], were also generated by PCR-mediated site-directed mutagenesis. Adenovirus expression vectors including Ad-HA-hPARPs were generated in accordance with the manufacturer's instructions (Agilent Technologies). Adenovirus vectors expressing short hairpin RNAs against human PARP1, PARP2, PARP13, and HDAC1 (Ad-hHSF1-KD and so on) were generated using oligonucleotides listed in Supplementary Table 1. Briefly, the oligonucleotides were annealed and inserted into pCR2.1-hU6 at a BamHI/HindIII site, and then the XhoI/HindIII fragment containing hU6-shRNA was inserted into a pShuttle vector (Stratagene)[32]. To generate an expression vector for His-tagged hPARP13 at the C terminus, the cDNA was inserted into pET21a (Novagen).

**Cell cultures and treatments**. HeLa (ATTC, CCL-2), HEK293 (ATCC, CRL-1573) and wild-type and HSF1-null MEF cells[31] were maintained at 37 °C in 5% $CO_2$ in Dulbecco's modified Eagle's medium (DMEM) (Gibco) containing 10% fetal bovine serum (FBS) (Sigma). HeLa cells were treated with 0.5 μM DOX (Enzo Life Sciences) for 16 h, 10 Gy ionizing radiation (IR) (MBR-1520R-4, Hitachi Power Solutions) and recovery for 1 h, 100 J m$^{-2}$ Ultraviolet (UV)-C (Airtech UV lamp A15436, Ultra-Violet Products UVX radiometer) and recovery for 1 h, heat shock (HS) at 42 °C for 30 min, 10 μM MG132 (Sigma-Aldrich) for 6 h, or 5 mM L-azetidine-2-carboxylic acid (AZC) (Tokyo Chemical Industry) for 6 h. Some cells were pretreated for 2 h with PARP inhibitor PJ34 (Enzo Life Sciences) (20 μM). Cells were also treated with a nuclear export inhibitor Leptomycin B (LMB) (Enzo Life Sciences) (20 nM). To knockdown HSF1, PARP1, PARP13, HDAC1, or PARP2, HeLa cells were infected with Ad-sh-hHSF1-KD, Ad-hPARP1-KD, Ad-hPARP13-KD, Ad-hHDAC1-KD, or Ad-hPARP2-KD ($1 × 10^7$ pfu/ml), respectively, for 2 h and maintained in normal medium for 70 h. To replace endogenous HSF1 with exogenous HSF1 or hHSF1-T20A, HeLa cells were infected with Ad-sh-hHSF1-KD ($1 × 10^7$ pfu/ml) for 2 h and maintained in normal medium for 22 h. They were then infected with Ad-hHSF1, Ad-hHSF1-T20A, or Ad-hHSF1-T20G ($2 × 10^6$ pfu/ml) for 2 h and maintained with normal medium for a further 46 h. Replacement of PARP1, PARP13, or HDAC1 with its mutant was performed in the same way.

Human mammary tumor MCF7 (RCB1904, RIKEN BRC) cells were maintained at 37 °C in 5% $CO_2$ in 10% FBS-containing DMEM. Human mammary tumor HCC1937 (CRL-2336, ATCC) and MDA-MB-436 (HTB-130, ATCC) cells were maintained in 10% FBS-containing RPMI 1640 (Gibco) and L-15 medium (Sigma), respectively. Mouse KB1P-G3 and KB1P-B11 cell lines, which were derived from a $BRCA1^{-/-}p53^{-/-}$ mammary tumor of FVB/N mice (a gift from Dr. Sven Rottenberg, the Netherlands)[55], were maintained in DMEM/F-12 (Gibco) containing 10% FBS, 50 μg/ml insulin (Sigma), 5 ng/ml epidermal growth factor (Life Technologies), and 5 ng/ml cholera toxin (Wako Chemicals USA, Inc.) in low oxygen conditions (3% $O_2$, 5% $CO_2$) at 37 °C. Replacement of endogenous HSF1 with hHSF1 or its mutant was performed as described above, except for the titers of infected viruses. For instance, mouse KB1P-G3 cells were infected with Ad-sh-mHSF1-KD ($5 × 10^7$ pfu/ml), and then infected with Ad-hHSF1, Ad-hHSF1-T20A, or Ad-hHSF1-T20G ($2 × 10^6$ pfu/ml).

**Antibodies**. We generated rabbit antisera against human PARP1 (αhPARP1-1, 1/1000) and PARP13 (αhPARP13-1, 1/1000 for WB and IF) by immunizing rabbits with bacterially expressed recombinant GST-hPARP1 (amino acids 1–373) and GST-hPARP13 (full-length), respectively. The following antibodies were also used: HSF1 (Millipore ABE1044, 1/1000), PARP1 (Santa Cruz sc25780, 1/100; Active Motif 39559 for ChIP, immunoprecipitation; αhPARP1-1 for ChIP and WB, 1/1000), PAR (Trevigen 4335-MC-100, 1/1000), acetylated lysine (Cell Signaling 9441S, 1/1000), HDAC1 (Abcam ab7028, 1/1000; Cell Signaling 5356, 1/1000), γH2AX (Millipore 05-636, 1/1000 for IF), 53BP1 (Cell Signaling 4937, 1/100 for IF), RAD51 (Abcam ab176458, 1/1000 for IF), ATM (Cell Signaling 2873, 1/1000),

ATR (Cell Signaling 2790, 1/1000), DNA-PK (Cell Signaling 4602, 1/1000), β-actin (Sigma-Aldrich A5441, 1/1000), HA (Nacalai Tesque HA124 for WB, 1/1000; Roche 3F10 for immunoprecipitation), and GFP (Nacalai Tesque GF200, 1/1000).

**Western blotting**. Cells were lysed with NP40 lysis buffer containing 1.0% NP40, 150 mM NaCl, 50 mM Tris-HCl (pH 8.0), and protease inhibitors (1 μg/ml leupeptin, 1 μg/ml pepstatin, and 1 mM phenylmethylsulfonyl fluoride) or RIPA lysis buffer containing 1.0% NP40, 150 mM NaCl, 50 mM Tris-HCl (pH 8.0), 0.5% sodium deoxycholate, 0.1% sodium dodecylsulfate (SDS), and protease inhibitors. After centrifugation, aliquots of the supernatant were subjected to SDS-polyacrylamide gel electrophoresis (SDS-PAGE). After transferring to a nitrocellulose membrane using Trans-Blot transfer cell (Bio-Rad), the membrane was blocked with 5% milk/phosphate-buffered saline (PBS) at room temperature (RT) for 1 h. Primary antibodies were diluted in 2% milk/PBS, and incubated at RT for 1 h or at 4 °C overnight. Alternatively, some antibodies including PARP1 antibody were diluted with PBS containing 0.1% Tween 20 (PBST). The membrane was washed with PBS for 5 min three times, followed by incubation of HRP-conjugated secondary antibodies (goat anti-rabbit IgG, Cappel 55689, 1/1000; goat anti-mouse IgG, Cappel 55563, 1/1000; goat anti-rat IgG, Jackson 112-035-003, 1/1000) in 2% milk/PBS at RT for 1 h. The membrane was washed three times with PBST, and chemiluminescent signals from ECL detection reagents (Amersham) were captured on an X-ray film (Super RX, Fujifilm).

**GST pull-down assay**. Recombinant GST-hHSF1, GST-hHSF2, GST-hHSF4 proteins[32] were expressed in *Escherichia coli* by incubating with 0.2 or 0.4 mM isopropyl β-D-1-thiogalactopyranoside (IPTG) at 37 °C for 2 h or at 25 °C for 10 h, and purified with Glutathione Sepharose 4B (GE Healthcare). Recombinant hPARP13-His protein was similarly expressed and purified by Ni Sepharose 6 Fast Flow (GE Healthcare). The purified GST and His-tagged fusion proteins were mixed in NT buffer (50 mM Tris-HCl pH 7.5, 150 mM NaCl, 1 mM EDTA, 0.5% NP40, 1 mM PMSF, 1 μg/ml leupeptin, 1 μg/ml pepstatin) for 1 h at 4 °C, and were then incubated with 20 μl glutathione-sepharose beads for 2 h at 4 °C. After the beads had been washed with NT buffer five times, the bound proteins were analyzed by western blotting.

**Immunoprecipitation**. Cells were lysed with RIPA lysis buffer. After centrifugation, the supernatant containing 5 mg protein was incubated with 5 μl of antiserum or 2 μg IgG antibody for HSF1, PARP1, PARP13, or HA-tag at 4 °C for 16 h, and mixed with 40 μl protein A-Sepharose beads (GE Healthcare) by rotating at 4 °C for 1 h. The complexes were washed five times with RIPA lysis buffer, and subjected to immunoblotting (Supplementary Fig. 9).

To detect auto-PARylation of PARP1, we performed a denaturing immunoprecipitation. Cells ($1 × 10^7$) were lysed with 100 μl of denaturing buffer (1% SDS, 5 mM EDTA, 10 mM β-mercaptoethanol), and were heated at 95 °C for 10 min. The denatured lysates were mixed with 0.9 ml of RIPA lysis buffer. They were incubated with 5 μl of antiserum for PARP1 at 4 °C for 16 h, mixed with 40 μl protein A-Sepharose beads at 4 °C for 1 h, and subjected to immunoblotting using PAR antibody. Acetylation of PARP1 was also detected by the denaturing immunoprecipitation.

**Immunofluorescence**. HeLa cells were cultured on glass coverslips in 6 cm dishes at 37 °C for 24 h. Untreated cells or cells treated with DOX for 6 or 16 h, were fixed with 4% paraformaldehyde in medium at RT for 10 min. They were washed with PBS, permeabilized for 10 min with PBS containing 0.2% Triton X-100, blocked in 2% milk/PBS for 30 min at RT, incubated with primary antibodies diluted in 2% milk/PBS for 1 h at RT, and then incubated with FITC-conjugated goat anti-rabbit IgG (1:200 dilution) (Cappel) or Alexa Fluor 568-conjugated goat anti-mouse IgG (1:200 dilution) (Molecular Probes). The cover slips were washed and mounted in a VECTASHIELD mounting medium with 4′,6-diamidino-2-phenylindole (DAPI; Vector Laboratories). Fluorescence images were captured using Axiovert 200 fluorescence microscope (Carl Zeiss) or LSM510 META laser scanning confocal microscope (Carl Zeiss).

**Microarray analysis**. HeLa cells were infected with Ad-sh-hHSF1-KD to knockdown endogenous HSF1 or with Ad-sh-SCR as a control, and then infected with Ad-hHSF1 or Ad-hHSF1-T20A as described above. Total RNA was prepared using RNeasy Mini Kit (Qiagen), and subjected to microarray analysis using a GeneChip Human Gene 1.0 ST Array in accordance with the manufacturer's instructions (Affymetrix). Gene expression data were analyzed using Partek Genomics Suite 6.5 (Partek). Fold-change of each mRNA in HSF1 knockdown cells was evaluated by normalizing to the mRNA level in scrambled-RNA-treated cells (HSF1-KD/SCR). Fold-change in hHSF1-T20A-reexpressed cells after HSF1 knockdown were evaluated by normalizing to the mRNA level in hHSF1-reexpressed cells (T20A-HA/HSF1-HA). To identify DNA damage-inducible genes in HeLa cells, cells were treated with 0.5 μM DOX for 16 h, and fold-changes of each mRNA level in DOX-treated cells were evaluated by normalizing to the mRNA level in untreated cells. All reactions were performed in triplicate with samples derived from three experiments.

**Assessment of mRNA**. Total RNA was extracted from HeLa cells using Trizol (Invitrogen), and first-strand cDNA was synthesized using avian myeloblastosis virus reverse transcriptase (AMV-RT) and oligo $(dT)_{20}$ according to the manufacturer's instructions (Invitrogen). RT-PCR was performed using primers summarized in Supplementary Table 2. Real-time quantitative PCR (qPCR) was performed using the StepOnePlus (Applied Biosystems) with Power SYBR Green PCR master mix (Applied Biosystems) according to the manufacturer's instructions. Primers used for qRT-PCR reactions are listed in Supplementary Table 3. Relative quantities of mRNAs were normalized against β-actin mRNA levels. All reactions were performed in triplicate with samples derived from three experiments.

**ChIP assay**. ChIP assay was performed using a kit in accordance with the manufacturer's instructions (EMD Millipore). The antibodies used for the ChIP assay are described above. Real-time quantitative PCR (qPCR) of ChIP-enriched DNAs was performed using the primers listed in Supplementary Table 4. Percentage input was determined by comparing the cycle threshold value of each sample to a standard curve generated from a 5-point serial dilution of genomic input, and compensated by values obtained using normal IgG. IgG-negative control immunoprecipitations for all sites yielded < 0.05% input. All reactions were performed in triplicate with samples derived from three experiments.

**ChIP-seq and data analysis**. HeLa cells were infected for 2 h with Ad-sh-SCR or Ad-sh-hPARP13-KD in normal medium for 22 h. These cells were then infected for 2 h with Ad-hHSF1 and Ad-hPARP1 ($2 × 10^6$ pfu/ml) and maintained with normal medium for 40 h, and were treated with Leptomycin B (20 nM) for 6 h. They were then fixed in 1% formaldehyde-containing medium at 37 °C for 10 min. Cells were washed twice with cold PBS, and suspended in 1.5 ml cell lysis buffer (LB1; 20 mM Tris-HCl (pH 7.5), 10 mM NaCl, 1 mM EDTA, 0.2% NP-40, 1 mM PMSF) on ice for 10 min. Nuclei were pelleted and suspended in 1.5 ml nuclei washing buffer (LB2; 20 mM Tris-HCl (pH 8.0), 200 mM NaCl, 1 mM EDTA, 0.5 mM EGTA, protease inhibitor cocktail) on ice for 10 min. Nuclei were pelleted and suspended in 1 ml sonication buffer (LB3; 20 mM Tris-HCl (pH 8.0), 150 mM NaCl, 1 mM EDTA, 0.5 mM EGTA, 1% Triton X-100, 0.1% Na-Deoxycholate, 0.1% SDS, protease inhibitor cocktail) at room temperature for 10 min, and then pelleted and resuspended in 400 μl sonication buffer on ice for 10 min. Resuspended chromatin was sonicated with the Sonifier 450 (Branson Ultrasonics) into fragmented DNA of ~200–300 bp, centrifuged at 15,000 rpm for 5 min, and transferred into new tubes. Fifty microliters of Dynabeads Protein A (Invitrogen) was washed twice with 5 mg/ml BSA in PBS, preincubated with 4 μg of PARP1 antibody (αhPARP1-1) at 4 °C for 3 h, and suspended in 100 μl sonication buffer. These beads were incubated at 4 °C overnight with sonicated chromatin, and were washed five times with RIPA wash buffer (50 mM HEPES-KOH (pH 7.4), 0.25 M LiCl, 1 mM EDTA, 0.5% Na-Deoxycholate, 1% NP-40) and once in TE50 buffer (50 mM Tris-HCl, 10 mM EDTA), and were then incubated in 200 μl EB buffer (TE50 buffer containing 1% SDS) at 65 °C for 20 min to immunoprecipitated materials. The eluted chromatin was de-crosslinked at 65 °C for 6 h, and was treated with RNase A at 50 °C for 1 h, and then with proteinase K at 50 °C overnight. DNA was extracted with phenol-chloroform, followed by ethanol precipitation, and was then purified using QIAquick PCR Purification Kit (QIAGEN). ChIP-seq libraries were prepared using NEBNext ChIP-Seq Library Prep Master Mix Set for Illumina (New England Biolabs), and were run on the HiSeq2000 sequencer (Illumina) to generate single-end 50-bp reads[29].

Sequenced reads obtained by ChIP-seq were mapped to the human genome (UCSC hg19) using Bowtie version 1.1.2[68], allowing two mismatches in the first 28 bases per read (-n2 option). We only considered uniquely mapped reads; redundantly mapped reads (reads starting exactly at the same 5′-sequence ends) were filtered out for further analysis. For peak calling and data visualization, we used DROMPA[69] (version 2.6.4 with 1-kbp bin). To compare multiple ChIP-seq data, ChIP and Input reads were both normalized with the total number of mapped reads. The identified peaks satisfied the following criteria: fold enrichment (ChIP/Input) >2.0, $P < 1 × 10^{-4}$, and normalized peak intensity >3.0.

**HRR and NHEJ assays**. An expression vector pCBASce, which encodes the rare-cutting I-SceI endonuclease from *Saccharomyces cerevisiae*, was transfected into HeLa-DRGFP cells (a gift from Dr. Junya Kobayashi, Kyoto, Japan)[51], in which a single copy of pDR-GFP reporter[50] is stably integrated into the genome, using Lipofectamine 2000 (Invitrogen). Repair of an I-SceI-induced DSB by HRR activated the expression of GFP. The numbers of GFP-positive cells and nuclei stained with DAPI were counted at 48 h after transfection using an Axiovert 200 fluorescence microscope (Carl Zeiss), and the percentage of GFP-positive cells out of 200 cells was calculated to evaluate HRR efficiency. The efficiency of NHEJ was similarly estimated by using HeLa cells, in which a pEJSSA reporter[52] is stably integrated into the genome (a gift from Dr. Yosef Shiloh, Israel)[53].

**Cells harboring reporter constructs**. To monitor the promoter activity of human *GADD34* gene, we amplified the DNA fragment of its upstream region (−1000 to +48) by PCR using genomic DNA of HeLa cells[41], and inserted it into ptk-galp3-luc at the HindIII/XhoI site[70]. An I-SceI recognition site was created just upstream of the XhoI site. This reporter plasmid pGADD34-Luc and pGADD34ΔHSE-Luc,

in which the HSE sequences were deleted (Fig. 6h), were co-transfected with pcDNA3.1-neo (Invitrogen) into HeLa cells using Lipofectamine 2000 (Invitrogen), and cells were grown in medium containing 1.5 mg/ml of G418. Fifteen colonies were picked for each cell line, HeLa-pGADD34-Luc or HeLa-pGADD34ΔHSE-Luc. Two clones containing stably integrated copies of each reporter construct were used for reporter analysis, and representative data are shown.

To monitor binding of PARP1 to the reporters, HeLa-pGADD34-Luc and HeLa-pGADD34ΔHSE-Luc cells were transfected with an I-SceI expression vector pCBASce for 24 h, and ChIP assay was performed using the primers in Supplementary Table 4. The GADD34-HSE primer set amplified the HSEs on both endogenous *GADD34* promoter and the pGADD34-Luc reporter. PARP1 was redistributed markedly at the LUC region in the reporter during I-SceI treatment.

**Comet assay**. A comet assay at neutral pH condition was performed using a CometAssay kit in accordance with the manufacturer's instructions (Trevigen, Gaithersburg, MD). Briefly, HeLa cells, in which endogenous HSF1 was substituted with ectopically expressed hHSF1 or its mutant, were treated with DOX for indicated periods to induce DNA damage. Trypsinized cells were suspended with PBS ($5 × 10^5$ cells/ml), mixed with LMAgarose at a ratio of 1:10, and poured onto CometSlide. After the gel hardened at 4 °C, the slide was immersed in Lysis Solution for 1 h, and then subjected to electrophoresis in neutral buffer. DNA was visualized by SYBR Gold, and was imaged using LSM510 META laser scanning confocal microscope (Carl Zeiss). DNA fluorescence intensity in tail and head, and the tail length were quantified in 25 cells, and the tail moment values are shown. To quantify constitutive DNA damage in BRCA1-null mouse mammary tumor cells, the percentage of DNA-damaged cells with DNA fluorescence in the tail are shown.

**Clonogenic assay**. To determine the effects of DOX on colony-forming capacity, HeLa cells, in which endogenous HSF1 was substituted with ectopically expressed hHSF1, its mutant, or β-galactosidase, were seeded on 60 mm dishes at low density. The next day, they were treated with 0.5 μM DOX for 16 h, washed with PBS three times, and then maintained in complete medium. After 7 to 14 days, the colonies were fixed with 4% paraformaldehyde/PBS for 10 min, stained with 2% Giemsa's Stain Solution (Nacalai Tesque), and counted using a stereomicroscope. Colony numbers relative to that of DOX-untreated, scrambled RNA-treated cells are shown as surviving fractions. Each experiment was done three times.

**Tumor growth in vivo**. We generated the pact-luc vector by substituting the HSV-thymidine kinase promoter of ptk-galp3-luc with the human β-actin promoter[70], and transfected it into HeLa and KB1P-G3 cells. Stable transformants, HeLa-luc and KB1P-G3-luc cell lines, were isolated in medium containing 1.5 mg/ml of G418, and the levels of luciferase were estimated using a luciferase assay. In these cells, endogenous HSF1 was substituted with ectopically expressed hHSF1, its mutant, or β-galactosidase as described above. These cells ($1 × 10^6$ cells) were suspended in 50 μl PBS, mixed with 50 μl Matrigel (BD Biosciences), and inoculated into the fourth left mammary fat pad of anesthetized 5-week-old female FVB/N mice (CLEA Japan, Inc.) using a 26-gauge needle. Four weeks after the inoculation, tumor growth was monitored by in vivo bioluminescence assay (IVIS Spectrum, Perkin-Elmer) with D-luciferin (150 mg/kg) (VivoGloTM Luciferin, Promega). All experimental protocols relating to these mice were reviewed and approved by the Committee for Ethics on Animal Experiments of Yamaguchi University Graduate School of Medicine.

**Statistical analysis**. Data were analyzed with Student's *t*-test or ANOVA. Asterisks in figures indicate significant differences ($P < 0.01$ or $0.05$). Error bars represent standard deviations (s.d.) for three or more independent experiments.

**Data availability**. Microarray data from this study have been deposited in the NCBI Gene Expression Omnibus (GEO) database under accession number GSE80535. PARP1 ChIP-seq data have been deposited in the Sequence Read Archive database under accession number SRP100596. All other data are available from the authors upon request.

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

## Acknowledgements

We are grateful to Drs K. Sugasawa and T. Oda for valuable comments, and Dr. N. Yokota for ChIP-seq data analysis. We thank Drs D. Cui and E. Ikeda for equipments, Dr. M. Okuda for advice on the comet assay, Drs J. Kobayashi, M. Jasin, and Y. Shiloh for plasmids and cells for HRR and NHEJ assays, and Drs A. Duarte, S. Rottenberg, T. Kondo, and H. Itoh for BRCA1 mutant cell lines. This work was supported by JSPS KAKENHI grant numbers 26116720, 15H04706, 16K07256 (to A.N. and M.F.), 15H05976, 15H02369 (to K.S.), Takeda Science Foundation Special Project Research (to A.N.), and The Yamaguchi University "Pump-Priming Program" (to A.N.).

## Author contributions

M.F. and A.N. designed the project; M.F., R.T. and K.A. performed the experiments; E.T. and M.F. performed microarray analysis; R.N., M.F. and K.S. performed and analyzed ChIP-seq experiments. M.F. and A.N. wrote the manuscript. All authors discussed the results and commented on the manuscript.

## Additional information

**Competing interests:** The authors declare no competing financial interests.

