## [Peer Review File · Nature Communications]

Reviewers' Comments:

Reviewer #1:

Remarks to the Author:

HSF1 mediates DNA Damage Response by Assisting with the Redistribution of PARP1

Fujimoto et al present a highly comprehensive set of biochemical and cellular experiments demonstrating the existence of a ternary complex between PARP1, PARP13 and HSF1.

They convincingly demonstrate that this ternary complex occurs in vivo (IP experiments); show that they are mediated by direct interaction through a series of in vitro experiments (pull-downs); by use of deletion constructs map the regions required for the interaction between HSF1 and PARP13; and create mutant versions of HSF1 that can no longer bind to PARP13. However, they do not characterise the PARP13 / PARP1 interaction in detail.

Doxorubicin, IR and UV treatment of cells, releases PARP1 from the PARP13/HSF1 complex, in an auto-PARylation dependent manner.

They go on to show that this ternary complex is A) found at the BCL11A locus and B) can modulate/suppress the constitutive expression of several genes, including GADD34 and GADD45A (Growth Arrest and DNA damage), by binding to HSE elements found within the promoter region of these genes.

In addition, they show that HDAC1 is a component of the HSF1-PARP13-PARP1 complex in unperturbed cells, but is lost when cells are treated with DOX; resulting in highly acetylated PARP1 (which is correlated with high levels of auto-PARylation).

Finally, they examine the effects of HSF1 knockdown + mutant addbacks on DNA repair and cell proliferation using a series of biochemical and cellular assays. They also present some iso-graft data in mice.

The manuscript is clearly of a high experimental standard; however, the authors at several points do tend overstate the importance (and validity) of their conclusions.

Most of the manuscript deals with the discovery of a ternary complex between PARP1, PARP13 and HSF1, and the presence of a quaternary complex with HDAC1; which all nicely hangs together, however, the later experiments (described and presented in Figures 6 and 7) appear to be 'bolt-on' experiments, and do not sit as well into the overall narrative.

Major points:

POINT 1

The title of the manuscript should ideally be altered – to better indicate the discovery of an HSF1-PARP13-PARP1 ternary complex. The ternary complex clearly has a role to play in regulating a subset of genes, but it is not a global phenomenon required for DDR, per se, as alluded to in the current title.

POINT 2

A link between PARP1 + HSF1 has been noted before in the scientific literature. The authors should take care to acknowledge prior publications, and to place their work into the appropriate context.

POINT 3

Page 9: "auto-PARylation of PARP1 regulates its dissociation from HSF1-PARP13 during DNA damage".

The authors demonstrate that release of PARP1 from the PARP13/HSF1 complex is dependent on its autoPARylation (Fig 2a,b,c), which can be triggered by treating cells with doxyrubicin, IR and UV (i.e, known DNA damaging agents) but prevented by treatment with the PARP inhibitor PJ34.

There is quite a wealth of literature that indicates that auto-PARylation of PARP1 is dependent on, and is triggered by, binding of the protein to either ssDNA nicks/gaps or double-strand breaks.

Moreover, once auto-PARylated, PARP1 is reported to lose its ability to bind to DNA – so the outstanding, and somewhat unaddressed, question is how is PARP1 retained (albeit transiently) at the three GADD34 loci probed in CHIP experiments (Figure 4 / Supplementary Figure 4) ?

Presumably the observed PARP1 redistribution is due to the presence of downstream damage to DNA near the HSE element? Do DNA-binding mutants of PARP1 prevent this redistribution event?

POINT 4

The authors include a summary statement at the end of each section of the manuscript, unfortunately these tend to slightly overstate or over-interpret the significance of each set of experiments. Ideally these should be individually revisited and reworded.

e.g. see page 11: 'These results demonstrated that the HSF1-PARP13-PARP1 ternary complex enhances the DNA-damage-induced gene expression'.

In fact, the statement made at the end of the preceding paragraph (page 10) is more suitable and more accurate:

'These results indicated that the ternary complex suppresses constitutive expression of a set of DNA-damage inducible genes'

POINT 5

Page 12: "We found transient redistribution of PARP1 throughout the GADD34 locus at 6 h after DOX treatment, which was accompanied by PARylation of chromatin".

The authors should support their statement by more clearly referring to their data at 12 hours after DOX treatment (shown) in Supplementary Figure 4b; which clearly shows the "transient" nature of the redistribution, as the ability to chip PARP1 within the GADD34 gene.

POINT 6

Acetylation of PARP1 (counteracted by HDAC1) appears to also play a role in release of the protein from the PARP13/HSF1 complex. The interesting question is left unaddressed, which comes first? Loss of inhibitory HDCA1 or auto-PARylation?

POINT 7

Clearly knockdown of HSF1 or mutant add-backs perturb cellular DNA damage processes (Figure 6); a phenomenon that has been reported previously (see cited Ref. 48, Li and Martinez, 2011).

The results presented in Figure 6 therefore represent an incremental, but useful advancement, showing that the two HSF1 mutants (which are unable to bind to PARP13) phenocopy cells that lack HSF1.

Li and Martinez reported that cells lacking functional HSF1 are unable to form 53BP1 nuclear foci – and thus are compromised in their ability to repair DNA damage. The statement “Thus, the ternary complex promotes the repair of DSBs, in part by enhancing the recruitment of gammaH2AX, RAD51, and 53BP1” is therefore somewhat misleading and should be removed (or rewritten).

What does effect does knockdown of PARP13 (the intermediary in the ternary complex) have on the DNA damage response in cells? This experiment would add an important control, and add credence to the authors proposed model.

POINT 8

“Taken together, these results demonstrated that the ternary complex promotes HRR and NHEJ by facilitating the redistribution of PARP1”.

Loss of HSF1 clearly perturbs DNA repair (see previous point) – and directly affects the accumulation of 53BP1 into nuclear foci; a process known to be required for efficient NHEJ/HR, and the choice of which repair pathway to engage. The authors therefore have not definitely demonstrated that the ternary complex, per se, promotes HR / NHEJ through the process described.

Minor points – typographical

Figure 3b; Cell cycle arrest (not arest). What do the numbers quoted correspond to? Should be in figure, rather than just in the figure legend.

Figure 4b; what does DKD represent (green bar in histogram), also Supp. Figure 4c (blue bar in histogram)

Scale bars in micrographs, where used, should be coloured white – they are currently illegible in red.

Page 21; “These cells are suffered” suffer

Page 21; “... believed that HSF1 maintains balance of proteome...” the balance

Page 21, “... On the other hand, malignant tumor cells are adapting to continuous DNA damage by the DDR” this sentence needs clarification as to its intended meaning

Minor points – other

Some data are present in both the main text and in the supplementary material e.g. Figure 4 panel C, and Supplementary Figure 4 panel C.

Reviewer #2:

Remarks to the Author:

The authors have discovered a new protein complex composed by HSF1-PARP1-PARP13 and HDAC1 that regulates DNA damage repair response. The authors have shown physical interaction between these proteins in two different human cell lines. They identified the protein domain involved in the interaction between PARP-13 and HSF1. They have shown that PARP1-HSF1 interaction depends on the presence of PARP13, a scaffold protein that constitutively binds HSF1. HSF1-PARP1-PARP13 bind to several gene promoters, some of them related to DNA damage repair such as GADD34. DNA damage induction by DOX induces Parylation of PARP-1, a posttranslational modification that mediates dissociation from HSF1, and redistributes PAPR1 into other sequences of GADD34, inducing its expression. The redistribution of PARP1 during DNA damage also depends on acetylation. They found that HDAC1 maintains a hypoacetylated PARP1 state under non-stress conditions keeping PARP1 within the HSF1-PAPR13 complex. In the absence of HDAC1 PARP1 is hyperacetylated and parylated, causing its redistribution and inducing the expression of DNA repair genes. PARP1 also contributes to the recruitment of DNA repair proteins into the DNA damage location such as H2AX and RAD51. The authors have also shown the biological importance of this ternary protein complex in BRCA1 null tumor growth using cell lines and mouse mutants. The experiments performed in this study are well conducted and the obtained results support the authors' conclusions. The manuscript represents an advance for the field, and clarifies how PARP1 and HSF1 are implicated in DNA damage response. However, a number of points should be clarified. These are listed below.

Major points:

- 1) It is proposed that HSF1 is acting as a repressor of DNA repair genes through binding to PARP1. In order to fully understand the mechanism that the authors propose, It is very important to show whether PARP1 activation by DNA damage promotes HSF1 release from the promoter genes or if PARP1 inhibits the transactivation activity of HSF1 under non-stressful conditions and once PARP1 is released from the complex, HSF1 activates transcription of those genes. In Figure 4c, the authors should also show if HSF1 is also redistributed into 1-2 sequences in the presence of DOX or if only PARP1 is redistributed to those sequences.
- 2) The authors show that HSF1 mutants i.e T20A do not interact with PARP13, and therefore do not interact with PARP1. This mutant shows increased expression of DNA repair genes as it is shown in the transcriptomic analysis (Figure 3b, i.e GADD34). In the presence of DOX during 2h, levels of GADD34 are also higher in the mutants than in the control (Figure 3e). However, this mutant shows increased DNA damage (Figure 6d). How do the authors reconcile these results?
- 3) The authors indicate that two HSF1 binding sites are present in PARP13 (Supplementary Figure 1f). Figure 1g shows that mutation of Z domain or WWE domain on PARP13 abolishes the interaction with HSF1 the same as the double mutation. Shouldn't HSF1 still interact with the single mutants, since another HSF1 site is still present in the protein, but be abolished in the double mutant?
- 4) In Figure 3e, the authors should include as a control of the experiment a PARP1 mutant (i.e HYA) to show that no alteration in GADD34 expression is observed.
- 5) The authors should consider to show HDAC1-PARP1 interaction. They could easily address this point by probing blots from figure 5d-e with HDAC1. How is HDAC1 regulated in the presence of DNA damage? Does HDAC1 release the complex upon DOX treatment allowing PARP1 to be acetylated and activated or it is HDAC1 activity inactivated but still bound to the complex?
- 6) The authors assume that the ternary complex HSF1-PAPR13-PARP1 operates in the cancer cell lines in the same manners as in Hela or HEK cells. However they should show that the ternary complex is also recapitulated in the cancer cell lines to conclude that the changes in cell number/cell survival is due to the alteration in the ternary complex composition.

Minor points:

- 1) Does the total cell lysate panel in Figure 1c include nuclear fraction? I feel Total and Nuclear panels are redundant giving the fact that PARP1 is only nuclear.
- 2) The authors should consider to include in Figure 1b an anti-His blot to show equal amounts of PAPR13 levels in the experiment

- 3) Figure 1a, the authors should describe the hPARP13S construct. It is missing from the text and figure legend.
- 4) Labeling of Figure 2g DOX in grey, it is white in the histogram.
- 5) Figure 3e labeling, Scr+GFP should be black dot.
- 6) Experiments conducted in Figure 1f do not provide substantial useful information. Changing the aminoacids to those present in HSF2 or HSF4 which do not bind PARP1 are not going to change the outcome. It would be interesting to modify residues in position 20 of HSF2 and HSF4 into T to check if there is new binding between HSF2 /HSF4 to PARP1.
- 7) Why is the CHIP-seq data performed in the presence of LMB?

Reviewer #3:

Remarks to the Author:

This manuscript describes a novel role for heat shock transcription factor 1 (HSF1) that is previously characterized and well known as a master regulator of the heat shock response, especially when cells and organisms are exposed to acute proteotoxic insults. Here, the authors report that HSF1 is involved in DNA damage response (DDR) and regulates genome integrity by recruiting PARP1 to DNA, which is a prerequisite for redistribution of PARP1 to the sites of DNA damage. In fact HSF1 forms a ternary complex with PARP1 and PARP13, which is required for localization of PARP1 on the DNA at multiple genomic loci, including promoter regions of DNA-damage responsive genes, such as GADD45A and GADD 34. A loss of HSF1 results in decreased expression of DDR genes and subsequently to increased DNA damage. The authors also show that this particular function of HSF1 enhances DNA stability and promotes tumor growth, as they shown in the case of specific mammary tumorigenesis, i.e. BRCA1-null mammary tumors, which are sensitive to PARP inhibitors. The topic of the study is extremely timely and important. The obtained results are of high quality and well controlled, and they strongly support the novel function of HSF1 and its impact on a specific type of mammary tumorigenesis. Below, I provide with a list of suggestions to improve the manuscript.

1. Language throughout the manuscript should be improved. Although the text describes the results quite clearly, it is in many parts very short and hard to comprehend. For instance, in the Abstract, the reader might have difficulties to understand what the ternary complex is and by which mechanisms the cells are protected from DNA damage. Tempus should be uniform in the Abstract. The experimental work included in the manuscript is truly impressive, but the overall understanding of the work would benefit from more detailed explanations of the experiments and results thereof. As a minor add to the Introduction: the authors could mention that HSF1 undergoes various post-translational modifications, including acetylation, which will come up later in the manuscript.

2. What is the histone chaperone mentioned on page 5?

3. The ChIP-seq results should be explained better. The authors could add a figure to the main results to show the distribution of PARP1 binding in the whole genome, e.g. the percentages of localization in the promoter regions, coding regions etc. This would more strongly support the conclusion that HSF1 recruits PARP1 to the promoters of various genes. Similarly, the figure legend for the supplementary data of ChIP-seq in S2b should be improved, e.g. what is indicated with a red bar?

4. HSF1 is generally thought to act as a trans-activator and not a repressor. Thus, it is confusing that in their analyses of microarray results, the authors directly focus on the genes that were upregulated upon HSF1 KD. Please, explain better.

5. How big is overlay of the genomic regions identified in ChIP-seq and genes whose expression changes using microarray analyses?

6. The authors could refer to Zelin & Freeman (2015) when they discuss HDAC1 and HSF1.
7. The authors could consider merging Figure 1A with Figure 2, and the rest of the panels of Figure 1 could be moved to Figure S1. This would allow a faster move to the functional role of the ternary complex, which is a very important finding.
8. What does NC1 stand for in Figures 2f and 2g. In the legend for Figure 2d, the square for DO should be white of the bars in the figure should be grey. In Figure 3e, check the pattern indicating SCR+GFP.
9. On page 13, "PARP1 was also highly acetylated by DOX treatment" could use a reference to the figure.
10. On page 16, a whole paragraph describes only supplementary data. These results are important for understanding the study and should be included in the main figure.
11. For consistency, please indicate the molecular weight markers in all blots. Loading controls for the input samples would be good to show in Western blots.

Point-By-Point Response

RE: Manuscript NCOMMS-17-08733

We addressed the reviewer's concerns as described below, and used red font for the portions of the manuscript that have been revised.

***** Reviewer #1 *****

Thank you for valuable comments. We addressed the reviewer's concerns as described below.

Major points:

1) POINT 1

The title of the manuscript should ideally be altered to better indicate the discovery of an HSF1-PARP13-PARP1 ternary complex. The ternary complex clearly has a role to play in regulating a subset of genes, but it is not a global phenomenon required for DDR, per se, as alluded to in the current title.

To meet this concern, we changed "HSF1 mediates DNA damage response" to "HSF1 regulates genome integrity". We would like to use a phrase "by assisting with the redistribution of PARP1". New title is as below:

"HSF1 regulates genome integrity by assisting with the redistribution of PARP1"

2) POINT 2

A link between PARP1 + HSF1 has been noted before in the scientific literature. The authors should take care to acknowledge prior publications, and to place their work into the appropriate context.

To our knowledge, this is the first work that shows the complex formation of HSF1 with PARP1.

On the other hand, it has been reported that PARP1 regulates transcription of inducible genes in response to stimuli. One example is the heat shock response, which is characterized by the induction of *HSP* expression. We cited two reviews and one original report, which showed the redistribution of PARP1 on *Drosophila HSP70* gene locus, in the introduction section. In addition, Quararhni et al. reported that PARP1 is associated with *HSP70* gene and activates its transcription during heat shock in human HeLa cells (Genes Dev. 20, 3324-3336, 2006). We added this reference in the introduction section.

It was also reported that PARP1 deficiency altered the expression of *HSP70* expression in mouse fibroblasts (Fossati et al. Biochem. Cell Biol. 84, 703-712, 2006). This report supports the speculation that "the ternary complex may modulate the heat shock response", which we discussed in the last paragraph of the discussion section. We added this report and Quararhni's report in the discussion section.

3) POINT 3

Page 9: "auto-PARylation of PARP1 regulates its dissociation from HSF1-PARP13 during DNA damage" The authors demonstrate that release of PARP1 from the PARP13/HSF1 complex is dependent on its autoPARylation (Fig 2a,b,c), which can be triggered by treating cells with doxyrubicin, IR and UV (i.e, known DNA damaging agents) but prevented by treatment with the

PARP inhibitor PJ34. There is quite a wealth of literature that indicates that auto-PARylation of PARP1 is dependent on, and is triggered by, binding of the protein to either ssDNA nicks/gaps or double-strand breaks. Moreover, once auto-PARylated, PARP1 is reported to lose its ability to bind to DNA; so the outstanding, and somewhat unaddressed, question is how is PARP1 retained (albeit transiently) at the three GADD34 loci probed in CHIP experiments (Figure 4 / Supplementary Figure 4) ?

Presumably the observed PARP1 redistribution is due to the presence of downstream damage to DNA near the HSE element? Do DNA-binding mutants of PARP1 prevent this redistribution event?

PARP1 has two zinc finger domains (Zn1 and Zn2), which recognize DNA breaks (Eustermann et al. *Mol. Cell* 2015, reference 42). We substituted endogenous PARP1 with HA-hPARP1 Δ Z1 (deletion of Zn1) or HA-hPARP1 Δ Z2, and showed that this substitution did not affect constitutive PARP1 occupancy in the *GADD34* promoter, and PARP1 redistribution into the *GADD34* locus and PARylation during DOX treatment in Supplementary Fig. 4f. HA-hPARP1 Δ Z1-2, which lacked both Zn1 and Zn2, bound to the *GADD34* promoter at a lower level in unstressed condition, but still redistributed into the gene locus during DOX treatment. These results suggest that PARP1 redistribution into the *GADD34* locus is not related with DNA damage recognition. We described these results and cited related references as below:

“We examined whether two zinc finger domains (Zn1 and Zn2) of PARP1, which recognize DNA breaks (Eustermann et al. *Mol. Cell* 2015, reference 42), is required for redistribution into the *GADD34* locus or not. Substitution of endogenous PARP1 with HA-hPARP1 Δ Z1 (deletion of Zn1) or HA-hPARP1 Δ Z2 did not affect constitutive PARP1 occupancy in the *GADD34* promoter, PARP1 redistribution into the *GADD34* locus during DOX treatment, and PARylation (Supplementary Fig. 4f). HA-hPARP1 Δ Z1-2, which lacked both Zn1 and Zn2, bound to the *GADD34* promoter at a lower level in unstressed condition, but still redistributed into the gene locus during DOX treatment. These results suggest that PARP1 redistribution into the *GADD34* locus is not related with DNA damage recognition (Kotova et al. *PNAS* 2010, reference 43).”

4) POINT 4

The authors include a summary statement at the end of each section of the manuscript, unfortunately these tend to slightly overstate or over-interpret the significance of each set of experiments. Ideally these should be individually revisited and reworded.

e.g. see page 11: “These results demonstrated that the HSF1-PARP13-PARP1 ternary complex enhances the DNA-damage-induced gene expression ”

In fact, the statement made at the end of the preceding paragraph (page 10) is more suitable and more accurate: “These results indicated that the ternary complex suppresses constitutive expression of a set of DNA-damage inducible genes ”

Thank you for pointing it. We understand that in a summary statement at the end of each section, we tend to overstate or over-interpret the significance of each set of experiments. We rewrote the sentence “These results demonstrated that the HSF1-PARP13-PARP1 ternary complex enhances the DNA-damage-induced gene expression” as below:

“These results demonstrated that HSF1-PARP13-PARP1 ternary complex suppresses constitutive expression of *GADD34* and enhances its induction during DNA damage, and suggested that PARP1 mediates repressive activity of HSF1 under unstressed conditions.”

We also revisited other summary statements and rewrote some of them as shown in red.

5) POINT 5

Page 12: *“We found transient redistribution of PARP1 throughout the GADD34 locus at 6 h after DOX treatment, which was accompanied by PARylation of chromatin ”*

The authors should support their statement by more clearly referring to their data at 12 hours after DOX treatment (shown) in Supplementary Figure 4b; which clearly shows the “transient” nature of the redistribution, as the ability to chip PARP1 within the GADD34 gene.

Thank you for pointing it. We rewrote the results as below:

“We found that PARP1 redistributed from the HSE to regions 1, 2, and 3 on the GADD34 locus at 6 h after DOX treatment, and then disappeared at 12 h (Supplementary Fig. 4b). The transient redistribution of PARP1 on this locus was accompanied by PARylation of chromatin (Supplementary Fig. 4b).”

6) POINT 6

Acetylation of PARP1 (counteracted by HDAC1) appears to also play a role in release of the protein from the PARP13/HSF1 complex. The interesting question is left unaddressed, which comes first? Loss of inhibitory HDCA1 or auto-PARylation?

Thank you for pointing it. In new Fig. 5e, we examined the interaction of HDAC1 with PARP1, and showed that the release of HDAC1 was associated with acetylation and auto-PARylation of PARP1. Furthermore, we showed in new Supplementary Fig. 5a,b that HDAC1 was released even from inactive hPARP1 mutants during DOX treatment. These results suggest that HDAC1 was released independently of auto-PARylation of PARP1. We modified the text to describe these results as below:

“During DOX treatment, PARP1 was acetylated and auto-PARylated (Fig. 5c,d) and HDAC1 was released from PARP1 (Fig. 5e). HDAC1 was released even from inactive hPARP1 mutants, suggesting a mechanism that was independent of auto-PARylation (Supplementary Fig. 5a,b).”

7) POINT 7

Clearly knockdown of HSF1 or mutant add-backs perturb cellular DNA damage processes (Figure 6); a phenomenon that has been reported previously (see cited Ref. 48, Li and Martinez, 2011). The results presented in Figure 6 therefore represent an incremental, but useful advancement, showing that the two HSF1 mutants (which are unable to bind to PARP13) phenocopy cells that lack HSF1.

Li and Martinez reported that cells lacking functional HSF1 are unable to form 53BP1 nuclear foci, and thus are compromised in their ability to repair DNA damage. The statement “Thus, the ternary complex promotes the repair of DSBs, in part by enhancing the recruitment of gammaH2AX, RAD51, and 53BP1” is therefore somewhat misleading and should be removed (or rewritten).

Thank you for careful suggestion. We added a sentence and cited a reference (Li and Martinez, 2011) as below in the result section “HSF1-PARPs promotes DNA repair”.

*“Furthermore, HSF1 deficiency results in impaired DNA repair (Li and Martinez *Radiat. Res.* 2011, reference 47) ”*

We also rewrote the summary sentence of the paragraph as below:

“Thus, the ternary complex facilitates the recruitment of γ H2AX, RAD51, and 53BP1, and promotes DNA repair.”

What does effect does knockdown of PARP13 (the intermediary in the ternary complex) have on the DNA damage response in cells? This experiment would add an important control, and add credence to the authors proposed model.

We showed in Supplementary Fig. 6h-j that PARP13 knockdown also resulted in impaired recruitment of γ H2AX, RAD51 and 53BP1. We added a sentence in the result section as below:

PARP13 knockdown also resulted in impaired recruitment of the same repair factors (Supplementary Fig. 6h-j).

8) POINT 8

“Taken together, these results demonstrated that the ternary complex promotes HRR and NHEJ by facilitating the redistribution of PARP1 ”

Loss of HSF1 clearly perturbs DNA repair (see previous point), and directly affects the accumulation of 53BP1 into nuclear foci; a process known to be required for efficient NHEJ/HR, and the choice of which repair pathway to engage. The authors therefore have not definitely demonstrated that the ternary complex, per se, promotes HR / NHEJ through the process described.

We rewrote the sentence as below:

“Taken together, these results demonstrated that the ternary complex facilitates the redistribution of PARP1 and promotes DNA repair including HRR and NHEJ.”

We also replaced the title of the result “HSF1-PARPs promotes HRR and NHEJ by facilitating the redistribution of PARP1” to “HSF1-PARP13 facilitates PARP1 redistribution to DNA lesions”.

Minor points, typographical:

1) *Figure 3b; Cell cycle arrest (not arrest). What do the numbers quoted correspond to? Should be in figure, rather than just in the figure legend.*

We correctly label “arrest” in Fig. 3b. We wrote “function”, “Enrichment p-value”, and “gene name” at the top of this figure.

2) *Figure 4b; what does DKD represent (green bar in histogram), also Supp. Figure 4c (blue bar in histogram)*

In the legends of Fig. 4b and Supplemental Fig. 4c, we added a sentence below:

“Double knockdown (DKD) indicates knockdown of both PARP1 and PARP13.”

3) *Scale bars in micrographs, where used, should be coloured white; they are currently illegible in red.*

We showed scale bars in white in Fig. 6a, Supplemental Fig. 1a, and Supplemental Fig. 6a,b.

4) *Page 21; “These cells are suffered”; suffer*

We correct it.

5) *Page 21; “believed that HSF1 maintains balance of proteome”; the balance*

We modified the sentence to “It is thought that HSF1 maintains the proteome balance”.

6) Page 21, “*On the other hand, malignant tumor cells are adapting to continuous DNA damage by the DDR*”; *this sentence needs clarification as to its intended meaning*

We rewrote it as below:

“On the other hand, malignant tumor cells need to adapt to some extent to continuous DNA damage in order to proliferate and the DDR plays a role in this process.”

Minor points, other:

1) *Some data are present in both the main text and in the supplementary material e.g. Figure 4 panel C, and Supplementary Figure 4 panel C.*

Because of space limitations, we only showed graphs of the HSF1 substitution experiment in main figure (Fig. 4c). The graphs of PARP1 and PARP13 knockdown experiment were shown in supplemental figure (Supplemental Fig. 4c). We presented other figures in a similar way.

***** Reviewer #2 *****

We are grateful to the reviewer for valuable comments. We revised the manuscript according to all of the referee’s comments below.

Major points:

1) *It is proposed that HSF1 is acting as a repressor of DNA repair genes through binding to PARP1. In order to fully understand the mechanism that the authors propose, It is very important to show whether PARP1 activation by DNA damage promotes HSF1 release from the promoter genes or if PARP1 inhibits the transactivation activity of HSF1 under non-stressful conditions and once PARP1 is released from the complex, HSF1 activates transcription of those genes. In Figure 4c, the authors should also show if HSF1 is also redistributed into 1-2 sequences in the presence of DOX or if only PARP1 is redistributed to those sequences.*

We showed in Supplementary Fig. 4d that HSF1 occupied the HSE before and after DOX treatment and did not redistribute to *GADD34* locus. Thus, PARP1 activation by DNA damage does not promote HSF1 release from the promoter. We described it as below:

“HSF1 occupied the HSE before and after DOX treatment and did not redistribute on the *GADD34* locus (Supplementary Fig. 4d).”

We also showed in Supplementary Fig. 3e that the basal level of *GADD34* expression was elevated, and the induced expression was markedly suppressed in PARP1 knockdown cells, like in HSF1 or PARP13 knockdown cells. In contrast, substitution with PARP1 mutants (HYA, AAA) did not alter the basal *GADD34* expression (see response to comment 4). These results suggested that PARP1 mediates a repressive activity of HSF1 under unstressed conditions.

We stated in the last paragraph of the results (pages 11-12) as below:

“These results demonstrate that HSF1-PARP13-PARP1 ternary complex suppresses constitutive expression of *GADD34* and enhances its induction during DNA damage, and suggest that PARP1 mediates repressive activity of HSF1 under unstressed conditions.”

Our observation did not support the possibility that PARP1 inhibits the transactivation activity of HSF1 under non-stressful conditions and once PARP1 is released from the complex, HSF1 activates transcription of those genes (as reviewer suggested).

2) *The authors show that HSF1 mutants i.e T20A do not interact with PARP13, and therefore do not interact with PARP1. This mutant shows increased expression of DNA repair genes as it is shown in the transcriptomic analysis (Figure 3b, i.e GADD34). In the presence of DOX during 2h, levels of GADD34 are also higher in the mutants than in the control (Figure 3e). However, this mutant shows increased DNA damage (Figure 6d). How do the authors reconcile these results?*

Thank you for pointing out this important issue. When DNA repair factors including homologous recombination repair (HRR) and NHEJ factors are induced, DNA damage should be suppressed. For example, the androgen receptor (AR) upregulates the expression of HR and NHEJ factors (BRCA1, RAD51AP1, RAD51C, RAD54L, DNAPKcs, etc), and thus promotes DNA repair during DNA damage (Goodwin et al, Cancer Discov. 2013, PMID: 24027197; Li et al, Science Signal. 2017, PMID: 28536297).

In HSF1-knockdown cells, many DNA damage-inducible genes, such as *GADD45A*, *GADD34*, *DDIT3*, *IL17R*, and *EPHA2*, were induced (Fig. 3b,d). These gene products are involved in the regulation of cell cycle, apoptosis, stress signaling and protein synthesis during DNA damage, but are not DNA repair factors (references 38-40). Therefore, DNA damage was not suppressed in cells expressing HSF1 mutants (Fig. 6d). In the results section, we explained more clearly functions of DNA damage-inducible genes upregulated in HSF1-knockdown cells as below:

“Gene ontology enrichment analysis showed that these included many DNA-damage-inducible genes, such as *GADD45A*, *GADD34*, *DDIT3*, *IL17R*, and *EPHA2* (Fig. 3b), whose products are not DNA repair factors but are involved in the regulation of cell cycle, apoptosis, stress signaling and protein synthesis during DNA damage³⁶⁻³⁸. ”

“Gene ontology enrichment analysis for the upregulated genes included many DNA damage-inducible genes, such as *GADD45A*, *GADD34*, *DDIT3*, *IL17R*, and *EPHA2* (Fig. 3b), whose products are not DNA repair factors but rather involved in the regulation of the cell cycle, apoptosis, stress signaling, and protein synthesis during DNA damage³⁸⁻⁴⁰. ”

Reference 39 (Hollander et al, Nat. Genet. 1999) was replaced with a suitable reference (Liebermann and Hoffman, J. Mol. Signal. 2008).

3) *The authors indicate that two HSF1 binding sites are present in PARP13 (Supplementary Figure 1f). Figure 1g shows that mutation of Z domain or WWE domain on PARP13 abolishes the interaction with HSF1 the same as the double mutation. Should HSF1 still interact with the single mutants, since another HSF1 site is still present in the protein, but be abolished in the double mutant?*

In vitro GST-pull down experiments showed that PARP13 has two HSF1-binding domains (Z and WWE domains) (Supplementary Fig. S1). However, as the reviewer pointed out, we did not detect the interaction of HSF1 with PARP13 mutant lacking one of two HSF1-binding domains in cell extracts (Fig. 1g). These results suggest that both Z and WWE domains in PARP13 are required for the interaction with HSF1 in vivo. We speculate that one of the two HSF1-binding domains can interact with HSF1, but is too weak to bind stably with HSF1 in vivo. We rewrote the text correctly as below:

“We overexpressed HA-hPARP13 mutants, which lacked one of two HSF1-binding regions (ΔZ and ΔWWE) or both regions (ΔZ - ΔWWE) (Supplementary Fig. 1g) in HEK293 cells, and performed a co-precipitation experiment. HSF1 was not co-precipitated with any of the three HA-hPARP13 mutants, which suggests that both the zinc finger and WWE domains in PARP13 are required for

stable interaction with HSF1 in vivo (Fig. 1g). PARP1 was co-precipitated with all the PARP13 mutants, which confirms that the zinc finger and WWE domains are not required for PARP1-PARP13 interaction (Fig. 1g). ”

Our assumption is supported by the fact that expression profiles of *GADD34* are similar in cells expressing HA-hPARP13 Δ Z, HA-hPARP13 Δ WWE, or HA-hPARP13 Δ Z- Δ WWE (Fig. 3f).

4) *In Figure 3e, the authors should include as a control of the experiment a PARP1 mutant (i.e HYA) to show that no alteration in GADD34 expression is observed.*

Thank you for pointing it. We examined the *GADD34* expression in cells expressing PARP1 mutants (HYA, AAA) during DOX treatment, and showed in Supplementary Fig. 3e that the basal levels of *GADD34* expression were unaltered, but the induced expression was markedly suppressed.

We stated as below:

“In cells expressing hPARP1 mutants (HYA, AAA), basal levels of *GADD34* expression were unaltered, but induced expression was markedly suppressed (Supplementary Fig. 3e). PARP1 knockdown elevated the basal level of *GADD34* expression like with knockdown of HSF1 or PARP13.”

5) *The authors should consider to show HDAC1-PARP1 interaction. They could easily address this point by probing blots from figure 5d-e with HDAC1. How is HDAC1 regulated in the presence of DNA damage? Does HDAC1 release the complex upon DOX treatment allowing PARP1 to be acetylated and activated or it is HDAC1 activity inactivated but still bound to the complex?*

Thank you for pointing it. In new Fig. 5d,e, we examined the expression and co-precipitation of HDAC1 in untreated and DOX-treated cells, and showed that HDAC1 was released from PARP1 as well as HSF1 during DOX treatment, and the release of HDAC1 was associated with acetylation and auto-PARYlation of PARP1. Furthermore, we showed in new Supplementary Fig. 5a,b that HDAC1 was released even from inactive hPARP1 mutants during DOX treatment. These results suggest that HDAC1 was released independently of auto-PARYlation of PARP1. We modified the text to describe these results as below:

“During DOX treatment, PARP1 was acetylated and auto-PARYlated (Fig. 5c,d) and HDAC1 was released from PARP1 (Fig. 5e). HDAC1 was released even from inactive hPARP1 mutants, suggesting a mechanism that was independent of auto-PARYlation (Supplementary Fig. 5a,b).”

6) *The authors assume that the ternary complex HSF1-PAPR13-PARP1 operates in the cancer cell lines in the same manners as in Hela or HEK cells. However they should show that the ternary complex is also recapitulated in the cancer cell lines to conclude that the changes in cell number/cell survival is due to the alteration in the ternary complex composition.*

We showed in a new supplementary Fig. 8d that HSF1 and PARP1 formed a complex through a scaffold protein PARP13 in human mammary tumor HCC1937 and MDA-MB-436 cells. We added a statement as below:

“ In these tumor cells, HSF1 and PARP1 formed a complex in a manner dependent on PARP13 (Supplementary Fig. 8d).”

Minor points:

1) *Does the total cell lysate panel in Figure 1c include nuclear fraction? I feel Total and Nuclear panels are redundant giving the fact that PARP1 is only nuclear.*

The total cell lysate included nuclear fraction in Fig. 1c. Therefore, we deleted data of immunoprecipitation using the total cell lysate.

2) *The authors should consider to include in Figure 1b an anti-His blot to show equal amounts of PAPR13 levels in the experiment.*

We added anti-His blots of input (PARP13-His) in Fig. 1b.

3) *Figure 1a, the authors should describe the hPARP13S construct. It is missing from the text and figure legend.*

We described about PARP13S in the text as below:

“We found that HSF1 interacted with PARP1, PARP13, and a truncated isoform PARP13S (reference 33) in cell extracts (Fig. 1a).”

These two isoforms were detected in other figures including Fig. 1c. Therefore, we stated in the results of Fig. 1c as below:

“PARP1 and PARP13 (full-length and truncated PARP13) interacted with HSF1 in nuclear fractions (Fig. 1c).”

4) *Labeling of Figure 2g DOX in grey, it is white in the histogram.*

Thank you for pointing it. We amended the color.

5) *Figure 3e labeling, Scr+GFP should be black dot.*

Thank you for pointing it. We replaced it with black dot.

6) *Experiments conducted in Figure 1f do not provide substantial useful information. Changing the amino acids to those present in HSF2 or HSF4 which do not bind PARP1 are not going to change the outcome. It would be interesting to modify residues in position 20 of HSF2 and HSF4 into T to check if there is new binding between HSF2 /HSF4 to PARP1.*

As the reviewer pointed it, we generated GST-hHSF2mu, in which the position 20 was replaced with T and position 33 with A, and performed GST-pull-down assay as described in Fig. 1b. However, PARP13-His protein was not pulled down with GST-hHSF2mu (Figure below). Similarly, PARP13-His protein was not pulled down with GST-hHSF4mu, in which the position 20 was replaced with T and position 33 with A (Figure below). Our observations suggest that the two residues of HSF are not sufficient to interact stably with PARP13, and PARP13 may contact in part to Thr20 at the helix α 1 (H1) in the winged helix-turn-helix motif as well as an adjacent Ala33.

Figure. Bacterially purified hPARP13-His was pulled down with purified GST, GST-hHSF1, GST-hHSF2, GST-hHSF4, GST-hHSF2mu, and GST-hHSF4mu, and subjected to immunoblotting. Input of PARP13-His was also shown at the bottom.

In Fig. 1f, we would like to show that the interaction of HSF1 with PARP13 is abolished by substitution of residues 20 (T) or 33 (A) with several amino acids. We rewrote the sentences as below.

“Substitution of Thr20 or Ala33 abolished the interaction without affecting DNA-binding activity in vitro (Supplementary Fig. 1c,d). In addition, substitution of these residues with other amino acids including those found in hHSF2 or hHSF4 also abolished the interaction (Fig. 1f). Thus, PARP13 may in part contact Thr20 at the helix α 1 (H1) in the winged helix-turn-helix motif as well as an adjacent Ala33 .”

7) *Why is the CHIP-seq data performed in the presence of LMB?*

Although other groups showed thousands of PARP1 peaks (references are shown in the Methods), we detected only a few ChIP-seq peaks of endogenous PARP1 using the same procedure. Because we found the ternary complex in this study, we tried to elevate concentration of PARP1 and HSF1 by overexpression. Because PARP13 is localized predominantly in the cytoplasm, we treated cells with LMB to elevate concentration of PARP13 (see Supplementary Fig. 1a). These treatments enabled us to detect hundreds of endogenous PARP1 peaks (Supplementary Fig. 2b). We added one sentence below in the figure legend of Supplementary Fig. 2b.

“PARP13 accumulated in the nucleus of LMB-treated cells (see Supplementary Fig. 1a).”

***** Reviewer #3 *****

Thank you for favorable and valuable comments. We tried to address the reviewer’s concerns as described below.

1) *Language throughout the manuscript should be improved.*

This manuscript was edited by a native English speaker.

Although the text describes the results quite clearly, it is in many parts very short and hard to comprehend. For instance, in the Abstract, the reader might have difficulties to understand what the ternary complex is and by which mechanisms the cells are protected from DNA damage. Tempus should be uniform in the Abstract.

Thank you for pointing them out. We rewrote the abstract to meet reviewer's suggestions.

The experimental work included in the manuscript is truly impressive, but the overall understanding of the work would benefit from more detailed explanations of the experiments and results thereof.

We revisited the text and modified many parts including the results of ChIP-seq and microarray analyses (these were noted in reviewer's comments) as shown in red.

As a minor add to the Introduction: the authors could mention that HSF1 undergoes various post-translational modifications, including acetylation, which will come up later in the manuscript.

We rewrote the third paragraph in the introduction section, and mentioned post-translational modification of HSF1.

2) *What is the histone chaperone mentioned on page 5?*

We modified the sentence as below:

“In fact, a small amount of the HSF1 trimer constitutively binds to nucleosomal DNA in complex with replication protein A and the histone chaperone FACT (facilitates chromatin transcription) (references 31, 32).”

3) *The ChIP-seq results should be explained better. The authors could add a figure to the main results to show the distribution of PARP1 binding in the whole genome, e.g. the percentages of localization in the promoter regions, coding regions etc. This would more strongly support the conclusion that HSF1 recruits PARP1 to the promoters of various genes. Similarly, the figure legend for the supplementary data of ChIP-seq in S2b should be improved, e.g. what is indicated with a red bar?*

Thank you for valuable suggestions. We explained ChIP-seq results in more detail as described below, by adding a result showing the distribution of PARP1 binding in the whole genome in the main Fig. 2f.

“To investigate the possibility that HSF1-PARP13 recruits PARP1 to the genome, we performed PARP1 chromatin immunoprecipitation sequencing (ChIP-seq) analysis using LMB-treated HeLa cells overexpressing hPARP1 and hHSF1 (Supplementary Fig. 2b). A total of 744 PARP1-binding peaks were identified; nearly 60% of peaks were found within promoters and bodies of annotated genes, and 38% of peaks were found in distal regions (Fig. 2f) (reference 37). Only 10 peaks were identified in these cells after PARP13 knockdown.”

We also amended the figure legend for Supplementary Fig. 2b and c.

4) *HSF1 is generally thought to act as a trans-activator and not a repressor. Thus, it is confusing that in their analyses of microarray results, the authors directly focus on the genes that were upregulated upon HSF1 KD. Please, explain better.*

Thank you for pointing it. We added a sentence and described as below:

“Although HSF1 is generally thought to act as an activator, HSF1 knockdown not only reduced expression of many genes but also increased the expression of a substantial number of other genes (Supplementary Fig. 3b). Among 79 upregulated genes in HSF1-knockdown cells, the expression of 71 genes (90%) was also elevated by substitution with hHSF1-T20A (Fig. 3a and Supplementary Fig. 3c).”

5) *How big is overlay of the genomic regions identified in ChIP-seq and genes whose expression changes using microarray analyses?*

Microarray analysis showed that the expression of 71 genes was up-regulated and expression of 20 genes was down-regulated in both HSF1 knockdown cells and cells expressing hHSF1-T20A (fold change $> +1.7$ or -1.7 ; $P < 0.05$) (Supplementary Fig. 3c). Furthermore, the expression of 590 genes was up-regulated and expression of 387 genes was down-regulated in both cells (fold change $> +1.3$ or -1.3 ; $P < 0.05$; $n=3$) (data not shown). On the other hand, we identified only 744 PARP1 peaks in the genome by ChIP-seq analysis. Among genes near the 744 PARP1 binding peaks, the expression of only 27 genes (3.6%) was altered. As the reviewer pointed, it is worth noticing that the expression of 3.6% of genes near the PARP1 binding peaks was altered in both HSF1 knockdown cells and cells expressing hHSF1-T20A, although we detected only a limited number of PARP1 peaks. We stated about the overlap in the legend of Supplementary Fig. 2c as below:

“Among genes near the 744 PARP1 binding peaks, the expression of 27 genes (3.6%) was altered in both HSF1 knockdown cells and cells expressing hHSF1-T20A (fold change $> +1.3$ or -1.3 ; $P < 0.05$) (see Supplementary Fig. 3c).”

We tried hard to identify PARP1 peaks. We elevated concentration of PARP1 and HSF1 by overexpression, and treated cells with LMB to elevate concentration of PARP13. Finally, we were able to identify 744 PARP1 peaks, but this number may not be sufficient to discuss about the overlap in detail in the result section (microarray results).

6) *The authors could refer to Zelin & Freeman (2015) when they discuss HDAC1 and HSF1.*

We added a reference (Zelin & Freeman *J. Mol. Biol.* 427, 1644-1654, 2015) in the result section “HDAC1 maintains HSF1-PARP13-PARP1 complex on gene promoters”.

7) *The authors could consider merging Figure 1A with Figure 2, and the rest of the panels of Figure 1 could be moved to Figure S1. This would allow a faster move to the functional role of the ternary complex, which is a very important finding.*

This is a valuable suggestion. We tried to move Fig. 1b-g to Supplementary Fig. 1. However, we have to make an additional supplementary figure because of many data. We feel that it is not good for readers to refer two supplementary figures at the beginning of the results. Furthermore, we cannot show Fig. 1a, Fig. 2, and an additional ChIP-seq-related data (suggested in comment 3) in one figure because space is limited. Therefore, we decided to leave Fig.1.

8) *What does NC1 stand for in Figures 2f and 2g. In the legend for Figure 2g, the square for DOX should be white of the bars in the figure should be grey. In Figure 3e, check the pattern indicating SCR+GFP.*

In the legend of new Fig. 2g, we stated that “ChIP-qPCR on the peak region (BCL11A) and negative control region (NC1) was performed ($n=3$)”.

In new Fig. 2h, we changed the color of the DOX square to gray.

In Fig. 3e, we change the color of the SCR+GFP circle to black.

9) *On page 13, “PARP1 was also highly acetylated by DOX treatment” could use a reference to the figure.*

We modified the sentence and rewrote it as below:

“During DOX treatment, PARP1 was acetylated and auto-PARylated (Fig. 5c,d) and HDAC1 was released from PARP1 (Fig. 5e). ”

10) *On page 16, a whole paragraph describes only supplementary data. These results are important for understanding the study and should be included in the main figure.*

On page 16, we described the results of Supplemental Fig. 6g,h. According to reviewer’s suggestion, we moved Supplemental Fig. 6g,h to new main figure Fig. 6h,i.

11) *For consistency, please indicate the molecular weight markers in all blots. Loading controls for the input samples would be good to show in Western blots.*

We indicated the molecular weight markers in all blots. We also showed a loading control (beta-actin blot) for the input samples in all new figures.

Reviewers' Comments:

Reviewer #1:

Remarks to the Author:

The manuscript has clearly improved as a result of the response to referees' comments.

But there still appears to be a fundamental 'disconnect' between what is clearly the two halves of the manuscript i.e. the roles of HSF1/PARP13/PARP1 in PART A) controlling gene expression and PART B) the role of the ternary complex in DNA repair, which still needs to be addressed.

AUTHORS RESPONSE to REFEREE #1

POINT 1: The title is still misleading; see original comment.

POINT 2: Acknowledged

POINT 3: The additional experiments demonstrating that PARP1 redistribution across the GADD34 locus is not dependant on ZnF1 / ZnF2 are useful, and a welcome addition to the manuscript. But please see additional comments below.

POINT 4: Acknowledged

POINT 5: Acknowledged

POINT 6: The additional experiments demonstrating that release of HDAC1 from PARP1 is independent of auto-PARylation are a useful addition to the manuscript.

POINT 7 + POINT 8: See comments below.

'PART A'

The authors have shown that HSF1/PARP13/PARP1 has a role to play in controlling the expression of certain genes, in response to cellular 'stress' (inc. DNA damage); with the presented data showing that PARP1 is tethered at HSE elements by the formation of an HSF1/PARP13/PARP1 complex; PARP1 is deacetylated by HDAC1 (in the context of this ternary complex) and thus presumably 'deactivated'; and that PARP1 redistribution across the GADD34 locus is independent of the DNA-binding activity of its ZnF1/ZnF2 domains.

Some important unanswered questions however still remain:

>> What is acetylating PARP1; is this acetylation a constitutive process or a DNA-damage phenomenon?

>> Is 'released' HDAC1 acetylated? (acetylation, and deactivation, of HDAC1 is already known to be modulated by the cellular response to 'stress'; see published literature)

Whilst answering these questions experimentally is probably outside of context of this manuscript, they should be explored and commented on in a revised Discussion.

Perhaps more importantly, there is still a mechanistic problem with the authors' model that has not been fully addressed [originally raised as POINT 3] – this is because the current understanding of PARP1 biology is at odds with their model; i.e. the poly-ADP-ribose activity of PARP1 is activated via its binding to damaged DNA; and that auto-PARylated PARP1 is essentially an inactive end-state.

>> It is still not clear -- in the context of the GADD34 locus – when PARP1 auto-PARYlation occurs – it this before or after its redistribution to downstream sites?

An alternative, compatible model is that PARP1 is first redistributed from the HSF1/PARP13 locus, then activated (no longer associated with HDAC1) at the downstream locus, resulting in PARYlation of the surrounding chromatin and then eventually RELEASED due to its own auto-PARYlation.

>> The authors state that auto-PARYlation of PARP1 “releases” it from HSF1/PARP13.

On a technical point, by careful re-examination of the revised manuscript, it is clear that Figure 2a and 2b do not actually show DISSOCIATION of PARP1 from HSF1-PARP13; more that auto-PARYlated PARP1 is UNABLE to bind to HSF1/PARP13. This is because the experiment has been performed by IP of PARP1 itself.

Figure 2c also shows that auto-PARYlated PARP1 is unable to bind to PARP13 and HSF1 (as this is an HSF1 IP).

Therefore, to confirm their statement that auto-PARYlated PARP1 is RELEASED from HSF1 / PARP13, the authors need to show that PARP1-HYA and PARP1-AAA (incapable of auto-PARYlation) mutants are RETAINED in an HSF1-IP (and not the PARP1 IP currently shown in Figure 2d).

This additional data would then add an important control and add credence to the authors' preferred model.

>> Whilst the authors have shown that PARP1-HYA and PARP1-AAA (auto-PARYlation mutants) are not capable of driving the increased expression of GADD34 in response to DNA damage, they haven't tested if PARP1-HYA and PARP1-AAA are actually redistributed across the GADD34 locus, or if they remain stuck at the initial HSE site (via HSF1/PARP13).

If PARP1-HYA / AAA aren't redistributed then this acts to confirm the authors' favoured model and hypothesis.

However, if PARP1-HYA /AAA are redistributed [TRUE] then an alternative model must be true.

'PART B'

The authors are still rather 'over-selling' the 'DNA-repair' angle of their manuscript. They should be more cautious in their data interpretation and the language used moderated accordingly.

Page 16, Line 256

'Thus, the ternary complex facilitates the recruitment of gammaH2AX, RAD51, and 53BP1, and promotes DNA repair'.

>> The authors have shown that loss of HSF1 (and PARP13) results in reduced levels of gammaH2AX signal (a post-translational modification; note H2AX is already there) and reduced recruitment of 53BP1 and RAD51. The language used needs to be careful and precise – as to not potentially misdirect or confuse the reader.

Page 16, Line 249

'PARP13 knockdown also resulted in impaired recruitment of the same repair factors (Supplementary Fig. 6h-j)'

>> NOTE: to formally demonstrate that HSF1 and PARP13 are working in the same cellular pathway, a full epistatic genetic analysis would be required.

Page 20, Line 306

“These results indicate that the ternary complex specifically supports growth of BRCA1-deficient mammary tumors partly by promoting DNA repair’

>> facilitating rather than ‘promoting’.

Page 23, Line 366

“We show that HSF1-PARP13-PARP1 ternary complex regulates DDR mechanisms”

Note that the authors have not demonstrated that the ternary complex regulates DDR. It does however clearly affect or perturb normal DDR.

>> NOTE: localisation of PARP1 close to an engineered DSB site will implicitly improve repair efficiency; through PAR-mediated signalling and downstream recruitment of DNA repair factors.

It is also worth noting that PARP1 may not be able to access certain chromatin states, via its DNA-binding domain (made up of ZnF1-Znf2), i.e. heterochromatin. By tethering PARP1 (via HSF1/PARP13) to these areas, you ensure that these regions can be readily remodelled, to allow DNA repair machinery access – this is a hypothesis supported by the authors’ data –and loss of this capability will result in reduce H2AX phosphorylation and hence downstream DNA repair enzyme recruitment (53BP1).

MINOR

To aid the reader it is important to always implicitly state what type of PARylation is being discussed.

> PAGE13, line 195, PARylation = chromatin PARylation

> PAGE 13, line 207, PARylation = chromatin PARylation

Reviewer #2:

Remarks to the Author:

The reviewed manuscript has significantly improved compared to its original submission. The authors have included new several experiments and addressed all previous concerns. The writing style has also improved. This is a high impact and rigorous work and I believe that this is now suitable for publication.

Reviewer #3:

Remarks to the Author:

The authors have very carefully revised the manuscript and I have no more concerns to raise.

Point-By-Point Response

RE: Manuscript NCOMMS-17-08733A

We addressed the reviewer's concerns as described below, and used red font for the portions of the manuscript that have been revised.

***** Reviewer #1 *****

Thank you for careful and valuable comments. We addressed the reviewer's concerns as described below.

The manuscript has clearly improved as a result of the response to Referees' comments.

But there still appears to be a fundamental "disconnect" between what is clearly the two halves of the manuscript i.e. the roles of HSF1/PARP13/PARP1 in PART A) controlling gene expression and PART B) the role of the ternary complex in DNA repair, which still needs to be addressed.

1) *AUTHORS RESPONSE to REFEREE #1*

POINT 1: The title is still misleading; see original comment.

POINT 2: Acknowledged

POINT 3: The additional experiments demonstrating that PARP1 redistribution across the GADD34 locus is not dependant on ZnF1 / ZnF2 are useful, and a welcome addition to the manuscript. But please see additional comments below.

POINT 4: Acknowledged

POINT 5: Acknowledged

POINT 6: The additional experiments demonstrating that release of HDAC1 from PARP1 is independent of auto-PARylation are a useful addition to the manuscript.

POINT 7 + POINT 8: See comments below.

Thank you for check our revision carefully. We here explain new title below.

The original comment of POINT 1 is: *"The title of the manuscript should ideally be altered to better indicate the discovery of an HSF1-PARP13-PARP1 ternary complex. The ternary complex clearly has a role to play in regulating a subset of genes, but it is not a global phenomenon required for DDR, per se, as alluded to in the current title."*

Original title:

"HSF1 mediates DNA damage response by assisting with the redistribution of PARP1"

Previous revision:

"HSF1 regulates genome integrity by assisting with the redistribution of PARP1"

We tried to indicate *"the discovery of an HSF1-PARP13-PARP1 ternary complex"* as the reviewer pointed. New title is as below:

"HSF1-PARP13-PARP1 complex facilitates the maintenance of genome integrity"

[PART A]

The authors have shown that HSF1/PARP13/PARP1 has a role to play in controlling the expression of certain genes, in response to cellular "stress" (inc. DNA damage); with the presented data showing that PARP1 is tethered at HSE elements by the formation of an HSF1/PARP13/PARP1 complex; PARP1 is deacetylated by HDAC1 (in the context of this ternary complex) and thus presumably "deactivated"; and that PARP1 redistribution across the GADD34 locus is independent of the DNA-binding activity of its ZnF1/ZnF2 domains.

2) *Some important unanswered questions however still remain:*

>> *What is acetylating PARP1; is this acetylation a constitutive process or a DNA-damage phenomenon?*

>> *Is "released" HDAC1 acetylated? (acetylation, and deactivation, of HDAC1 is already known to be modulated by the cellular response to "stress"; see published literature)*

Whilst answering these questions experimentally is probably outside of context of this manuscript, they should be explored and commented on in a revised Discussion.

We rewrote a part of the third paragraph in the discussion as below:

"PARP1 dissociates from PARP13 by DNA damage-induced auto-PARylation (Figs. 2 and 4). HDAC1 maintains the interaction by inactivating PARP1 through deacetylation, and DNA damage-induced dissociation of PARP1 from PARP13 is associated with elevated acetylation levels of PARP1 (Fig. 5). It is known that PARP1 is acetylated by the acetyltransferases p300/CBP and PCAF and deacetylated by a number of deacetylases including SIRT1 and HDAC1 (Krishnakumar and Kraus, 2010; reference 3). Furthermore, HDAC1 acetylation is rapidly induced under various stress conditions and increased acetylation of HDAC1 reduces its deacetylase activity (Yan et al, 2015; new reference 61). Although PARP1 is activated by binding to damaged DNA, it is also activated by acetylation during stress in a manner that is independent of DNA damage (Rajamohan et al, 2009; new reference 62). Our observations suggest that it is first activated in the ternary complex through an acetylation-mediated mechanism and is then released from that. Activation and auto-PARylation of PARP1 result in its release from chromatin, but modestly modified PARP1 may retain its association with chromatin (Krishnakumar and Kraus, 2010; reference 3). Detailed mechanism of PARP1 dissociation from HSF1-PARP13 and its redistribution during DNA damage will be uncovered in future."

I pull out the portions, which contain answers to the reviewer's questions.

What is acetylating PARP1;

"It is known that PARP1 is acetylated by the acetyltransferases p300/CBP and PCAF and deacetylated by a number of deacetylases including SIRT1 and HDAC1 (Krishnakumar and Kraus, 2010; reference 3)."

Is this acetylation a constitutive process or a DNA-damage phenomenon?;

"DNA damage-induced dissociation of PARP1 from PARP13 is associated with elevated acetylation levels of PARP1 (Fig. 5)"

Is "released" HDAC1 acetylated?;

"Furthermore, HDAC1 acetylation is rapidly induced under various stress conditions and increased acetylation of HDAC1 reduces its deacetylase activity (Yan et al, 2015; reference 61)."

3) *Perhaps more importantly, there is still a mechanistic problem with the authors' model that has not been fully addressed [originally raised as POINT 3]; this is because the current understanding of PARP1 biology is at odds with their model; i.e. the poly-ADP-ribose activity of PARP1 is activated via its binding to damaged DNA; and that auto-PARylated PARP1 is essentially an inactive end-state.*

We understand the reviewer's concern and discussed about them in the text as shown above (response to comment #2). I pull out an important portion.

“Activation and auto-PARylation of PARP1 result in its release from chromatin, but modestly modified PARP1 may retain its association with chromatin (Krishnakumar and Kraus, 2010; reference 3). Detailed mechanism of PARP1 dissociation from HSF1-PARP13 and its redistribution during DNA damage will be uncovered in future.”

>> It is still not clear -- in the context of the GADD34 locus--when PARP1 auto-PARylation occurs; it this before or after its redistribution to downstream sites?

An alternative, compatible model is that PARP1 is first redistributed from the HSF1/PARP13 locus, then activated (no longer associated with HDAC1) at the downstream locus, resulting in PARylation of the surrounding chromatin and then eventually RELEASED due to its own auto-PARylation.

Thank you for mentioning an alternative model. It seems that additional two results shown below (new Supplementary Fig. 2a and Fig. 4d) do not support this alternative model. Please see responses to the reviewer's comments below.

>> The authors state that auto-PARylation of PARP1 "releases" it from HSF1/PARP13.

On a technical point, by careful re-examination of the revised manuscript, it is clear that Figure 2a and 2b do not actually show DISSOCIATION of PARP1 from HSF1-PARP13; more that auto-PARylated PARP1 is UNABLE to bind to HSF1/PARP13. This is because the experiment has been performed by IP of PARP1 itself.

Figure 2c also shows that auto-PARylated PARP1 is unable to bind to PARP13 and HSF1 (as this is an HSF1 IP).

Therefore, to confirm their statement that auto-PARylated PARP1 is RELEASED from HSF1 / PARP13, the authors need to show that PARP1-HYA and PARP1-AAA (incapable of auto-PARylation) mutants are RETAINED in an HSF1-IP (and not the PARP1 IP currently shown in Figure 2d).

This additional data would then add an important control and add credence to the authors' preferred model.

Thank you for carefully reviewing our results. Immunoprecipitation using HSF1 antibody confirmed that the complex of HSF1-PARP13 with PARP1-HYA and PARP1-AAA (incapable of auto-PARylation) mutants is retained during DOX treatment. We added this data in new Supplementary Fig. 2a.

4) *>> Whilst the authors have shown that PARP1-HYA and PARP1-AAA (auto-PARylation mutants) are not capable of driving the increased expression of GADD34 in response to DNA damage, they haven't tested if PARP1-HYA and PARP1-AAA are actually redistributed across the GADD34 locus, or if they remain stuck at the initial HSE site (via HSF1/PARP13).*

If PARP1-HYA / AAA aren't redistributed then this acts to confirm the authors' favoured model and hypothesis.

However, if PARP1-HYA /AAA are redistributed [TRUE] then an alternative model must be true.

Thank you for pointing it out. We showed in new Fig. 4d that HA-hPARP1-HYA and HA-hPARP1-AAA remained binding to the HSE and did not redistributed across the *GADD34* locus during DOX treatment. This result supports our model.

We described this result in the result section as below:

“When endogenous PARP1 was substituted with HA-hPARP1-HYA and HA-hPARP1-AAA, they remained binding to the HSE and did not redistributed across the *GADD34* locus during DOX treatment (Fig. 4d).”

[PART B]

The authors are still rather “over-selling” the “DNA-repair” angle of their manuscript. They should be more cautious in their data interpretation and the language used moderated accordingly.

5) *Page 16, Line 256*

”Thus, the ternary complex facilitates the recruitment of gammaH2AX, RAD51, and 53BP1, and promotes DNA repair”

>> The authors have shown that loss of HSF1 (and PARP13) results in reduced levels of gammaH2AX signal (a post-translational modification; note H2AX is already there) and reduced recruitment of 53BP1 and RAD51. The language used needs to be careful and precise; as to not potentially misdirect or confuse the reader.

We correctly stated as below:

“Thus, loss of HSF1 or PARP13 results in reduced levels of γ H2AX signal, reduced recruitment of 53BP1 and RAD51, and impairment of DNA repair.”

6) *Page 16, Line 249*

”PARP13 knockdown also resulted in impaired recruitment of the same repair factors (Supplementary Fig. 6h-j)”

>> NOTE: to formally demonstrate that HSF1 and PARP13 are working in the same cellular pathway, a full epistatic genetic analysis would be required.

We correctly stated as below:

“The signal intensity of γ H2AX and number of RAD51 and 53BP1 foci after DOX treatment were also reduced in PARP13 knockdown cells.”

7) *Page 20, Line 306*

”These results indicate that the ternary complex specifically supports growth of BRCA1-deficient mammary tumors partly by promoting DNA repair”

>> facilitating rather than ”promoting”.

We amended it.

8) *Page 23, Line 366*

"We show that HSF1-PARP13-PARP1 ternary complex regulates DDR mechanisms"

Note that the authors have not demonstrated that the ternary complex regulates DDR. It does however clearly affect or perturb normal DDR.

We changed the statement to "we show that HSF1-PARP13-PARP1 ternary complex affects the DDR".

9) >> *NOTE: localisation of PARP1 close to an engineered DSB site will implicitly improve repair efficiency; through PAR-mediated signalling and downstream recruitment of DNA repair factors.*

Thank you for this valuable note. According to this note, we modified a summary sentence of the result section "Taken together, these results demonstrated that the ternary complex facilitates the redistribution of PARP1 and promotes DNA repair including HRR and NHEJ" (old Page 17, Line 268) as below:

"These results demonstrate that HSF1-PARP13 facilitates redistribution of PARP1 close to an engineered DSB site and improves DNA repair efficiency, probably through PAR-mediated signaling and downstream recruitment of DNA repair factors." (new Page 17-18)

10) *It is also worth noting that PARP1 may not be able to access certain chromatin states, via its DNA-binding domain (made up of ZnF1-Znf2), i.e. heterochromatin. By tethering PARP1 (via HSF1/PARP13) to these areas, you ensure that these regions can be readily remodelled,; this is a hypothesis supported by the authors' data and loss of this capability will result in reduce H2AX phosphorylation and hence downstream DNA repair enzyme recruitment (53BP1).*

Thank you for this valuable note. According to this note, we added sentences in the second paragraph of the discussion (Page 21) as below:

"PARP1 binding at these sites is dependent on HSF1 and PARP13, suggesting that HSF1-PARP13 recruits PARP1 to HSF1-binding regions, which are widely distributed in the genome (references 27-30). Although PARP1 may not be able to access certain chromatin states via its DNA-binding domain (made up of zinc finger domains) (Kotova et al, 2010; reference 43), HSF1-PARP13 could tether PARP1 to these areas to remodel chromatin."

11) *MINOR*

To aid the reader it is important to always implicitly state what type of PARylation is being discussed.

> *PAGE13, line 195, PARylation = chromatin PARylation*

> *PAGE 13, line 207, PARylation = chromatin PARylation*

We amended them.

Reviewers' Comments:

Reviewer #1:

Remarks to the Author:

With the additional experiments and amendments to the text, I believe the manuscript is now suitable for publication.

Point-By-Point Response

RE: Manuscript NCOMMS-17-08733B

REVIEWERS' COMMENTS:

Reviewer #1 (Remarks to the Author):

With the additional experiments and amendments to the text, I believe the manuscript is now suitable for publication.

Thank you for careful review of our manuscript.

In the revised version, we also changed a title to the editor's suggested title: The HSF1-PARP13-PARP1 complex facilitates DNA repair and promotes mammary tumorigenesis.